# MULTI-HYPOTHESIS 3D HUMAN POSE ESTIMATION METRICS FAVOR MISCALIBRATED DISTRIBUTIONS

## ABSTRACT

Due to depth ambiguities and occlusions, lifting 2D poses to 3D is a highly ill-posed problem. Well-calibrated distributions of possible poses can make these ambiguities explicit and preserve the resulting uncertainty for downstream tasks. This study shows that previous attempts, which account for these ambiguities via multiple hypotheses generation, produce miscalibrated distributions. We identify that miscalibration can be attributed to the use of sample-based metrics such as minMPJPE. In a series of simulations, we show that minimizing minMPJPE, as commonly done, should converge to the correct mean prediction. However, it fails to correctly capture the uncertainty, thus resulting in a miscalibrated distribution. To mitigate this problem, we propose an accurate and well-calibrated model called Conditional Graph Normalizing Flow (cGNFs). Our model is structured such that a single cGNF can estimate both conditional and marginal densities within the same model – effectively solving a zero-shot density estimation problem. We evaluate cGNF on the Human 3.6M dataset and show that cGNF provides a well-calibrated distribution estimate while being close to state-of-the-art in terms of overall minMPJPE. Furthermore, cGNF outperforms previous methods on occluded joints while remaining well-calibrated [1].

NEW

## 1 INTRODUCTION

The task of estimating the 3D human pose from 2D images is a classical problem in computer vision and has received significant attention over the years (Agarwal & Triggs, 2004; Mori & Malik, 2006; Bo et al., 2008). With the advent of deep learning, various approaches have been applied to this problem with many of them achieving impressive results (Martinez et al., 2017; Pavlakos et al., 2016; 2018; Zhao et al., 2019; Zou & Tang, 2021). However, the task of 3D pose estimation from 2D images is highly ill-posed: A single 2D joint can often be associated with multiple 3D positions, and due to occlusions, many joints can be entirely missing from the image. While many previous studies still estimate one single solution for each image (Martinez et al., 2017; Pavlakos et al., 2017; Sun et al., 2017; Zhao et al., 2019; Zhang et al., 2021), some attempts have been made to generate multiple hypotheses to account for these ambiguities (Li & Lee, 2019; Sharma et al., 2019; Biggs et al., 2020; Oikarinen et al., 2020; Li & Lee, 2020; Kolotouros et al., 2021; Wehrbein et al., 2021). Many of these approaches rely on estimating the conditional distribution of 3D poses given the 2D observation implicitly through sample-based methods. Since direct likelihood estimation in sample-based methods is usually not feasible, different sample-based evaluation metrics have become popular. As a result, the field's focus has been on the quality of individual samples with respect to the ground truth and not the quality of the probability distribution of 3D poses itself.

FIX

In this study, we show that common sample-based metrics in lifting, such as mean per joint position error, encourage overconfident distributions rather than correct estimates of the true distribution. As a result, they do not guarantee that the estimated density of 3D poses is a faithful representation of the underlying data distribution and its ambiguities. As a consequence, their predicted uncertainty cannot be trusted in downstream decisions, which would be one of the key benefits of a probabilistic model (Fig. 1).

NEW

In a series of experiments, we show that a probabilistic lifting model trained with likelihood provides a higher-quality estimated distribution. First, we evaluate the distributions learned by minimizing

---

[1] Code and pretrained model weights are available at `https://github.com/XXXX`.

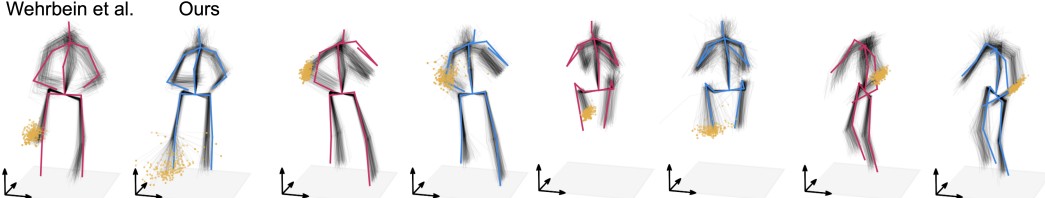

Figure 1: Examples showcasing the consequences of an overconfident distributions vs.our well-calibrated distribution. Ground truth marked with colored poses. Uses artificial 2D keypoint failures.

minMPJPE instead of negative log-likelihood (NLL) observing that, although minMPJPE optimal distributions have a good mean they are not well-calibrated. Next, we use the SimpleBaseline (Martinez et al., 2017) lifting model with a simple Gaussian noise model on Human3.6M to demonstrate that a model optimized for NLL is well-calibrated but underperforms on minMPJPE. The same model optimized for minMPJPE performs well in that metric but turns out to be miscalibrated. To balance this trade-off, we propose an interpretable evaluation strategy that allows comparing sample-based methods, while retaining calibration. Finally, we introduce a novel method to learn the distribution of 3D poses conditioned on the available 2D keypoint positions. To that end, we propose a Conditional Graph Normalizing Flow (cGNF). Unlike previous methods, cGNF does not require training a separate model for the prior and posterior. Thus, our model does not require an adversarial loss term, as opposed to Wehrbein et al. (2021). By evaluating the cGNF's performance on the Human 3.6M dataset (Ionescu et al., 2014), we show that, in contrast to previous methods, our model is well-calibrated while being close to state-of-the-art in terms of overall minMPJPE, and that it significantly outperforms prior work on occluded joints.

## 2  RELATED WORK

**Lifting Models**  Estimating the human 3D pose from a 2D image is an active research area (Pavlakos et al., 2016; Martinez et al., 2017; Zhao et al., 2019; Wu et al., 2022). An effective approach is to decouple 2D keypoint detection from 3D pose estimation (Martinez et al., 2017). First, the 2D keypoints are estimated from the image using a 2D keypoint detector, then a lifting model uses just these keypoints to obtain a 3D pose estimate. Since the task of estimating a 3D pose from 2D data is a highly ill-posed problem, approaches have been proposed to estimate multiple hypotheses (Li & Lee, 2019; Sharma et al., 2019; Oikarinen et al., 2020; Kolotouros et al., 2021; Li et al., 2021; Wehrbein et al., 2021). However, these approaches i) do not explicitly account for occluded or missing keypoints and ii) do not consider the calibration of the estimated densities. Wehrbein et al. (2021) incorporate a Normalizing Flow (Tabak, 2000) architecture to model the well-defined 3D to 2D projection and exploit the invertible nature of Normalizing Flows to obtain 2D to 3D estimates. Albeit structured as a Normalizing Flow it is not trained as a probabilistic model. Instead, the authors optimize the model by minimizing a set of cost functions. All in some form depend on the distance of hypotheses to the ground truth. In addition, they utilize an adversarial loss to improve the quality of the hypotheses. The proposed model achieves high performance on popular metrics in multi-hypothesis pose estimation, which are all sample-based distance measures rather than distribution-based metrics. Sharma et al. (2019) introduces a conditional variational autoencoder architecture with an ordinal ranking to disambiguate depth. Similarly to Wehrbein et al. (2021), the authors additionally optimize the poses on sample-based reconstruction metrics and report performance on sample-based metrics only. Oikarinen et al. (2020) utilize a graph-based approach to construct a mixture density network of Gaussian distributions. Kolotouros et al. (2021) use a volume-preserving normalizing flow model based on the GLOW architecture (Kingma & Dhariwal, 2018).  NEW

**Sample-Based Metrics in Pose Estimation**  The most widely used metric in pose estimation is the mean per joint position error (MPJPE) (Wang et al., 2021). It is defined as the mean Euclidean distance between the $K$ ground truth joint positions $\boldsymbol{X} \in \mathbb{R}^{K \times 3}$ and the predicted joint positions $\hat{\boldsymbol{X}} \in \mathbb{R}^{K \times 3}$. Multi-hypothesis pose estimation considers $N$ hypotheses of positions $\hat{\boldsymbol{\mathsf{X}}} \in \mathbb{R}^{N \times K \times 3}$

and adapts the error to consider the hypothesis closest to the ground truth (Jahangiri & Yuille, 2017).

$$\text{minMPJPE}(\hat{\mathbf{X}}, \boldsymbol{X}) = \min_n \frac{1}{K} \sum_k^K \left\| \hat{\mathbf{X}}_{n,k} - \boldsymbol{X}_k \right\|_2$$

In this work, we refer to this minimum version of the MPJPE as minMPJPE. Procrustes-Aligned MPJPE (PA-MPJPE) is a variation on MPJPE which first aligns the test pose to the ground truth pose. The percentage of correct keypoints (PCK) (Toshev & Szegedy, 2013; Tompson et al., 2014; Mehta et al., 2016) is another widely accepted metric in pose estimation which measures the percentage of keypoints in a circle of 150mm around the ground truth in terms of minMPJPE. Correct pose score (CPS) proposed by Wandt et al. (2021) considers a pose to be correct if all the keypoints are within a radius $r \in [0\,\text{mm}, 300\,\text{mm}]$ of the ground-truth in terms of minMPJPE. CPS is defined as the area under the curve of percentage correct poses and $r$.

NEW

**Calibration** is an important property of a probabilistic model measuring a model's ability to correctly reflect the uncertainty in the data. Thus, the confidence of an event assigned by a well-calibrated model should be equal to the true probability of the event (Brier, 1950). Guo et al. (2017) show that calibration of densities is especially important in the field of deep learning, where different architecture choices have been shown to lead to miscalibrated. Naeini et al. (2015) propose to measure the expected calibration error (ECE) metric which approximates the expectation of the absolute difference between the predicted probability and the true probability.

FIX

$$\text{ECE} = \frac{1}{N} \sum_{n=1}^N |\hat{\boldsymbol{p}}_n - \boldsymbol{p}_n| \tag{1}$$

The lower the ECE the better the calibration of the distribution. A model which predicts the same probability for all samples has an ECE of 0.5, whereas a perfectly calibrated model has ECE = 0. DeGroot & Fienberg (1983) and Niculescu-Mizil & Caruana (2005) provide a visual representation of calibration using **reliability diagrams**. They display the calibration curve, which is a function of confidence against the true probability. If the calibration curve is an identity function then the model is perfectly calibrated.

## 3 OBSERVING MISCALIBRATION

In this section, we demonstrate that the current state-of-the-art lifting models are not well-calibrated. We consider two of the latest methods: Sharma et al. (2019) and Wehrbein et al. (2021). We compute the ECE for the two models and visualize their reliability diagrams (Fig. 2a).

### 3.1 QUANTILE CALIBRATION FOR POSE ESTIMATION

Quantile calibration (Song et al., 2019) defines a perfectly calibrated distribution as one for which ground-truth values $\mathbf{X}^*$ fall within the $q$-th quantile $q\%$ of the time. However, for high dimensions estimating whether a point is contained within a given quantile is non-trivial. We, therefore, propose to simplify the problem by projecting to the univariate space of squared errors $\varepsilon$ from the median $\tilde{\mathbf{X}}$ of $N$ hypotheses $\hat{\mathbf{X}}$ conditioned on 2D poses $\mathbf{C}$ with $K$ keypoints. We then compute ECE in the space of $\varepsilon$ over the set of quantiles $\mathcal{Q} \in [0, 1]$ (Algorithm 1). As a measurement of central tendency we choose the median statistic, which is more robust to outlier samples. However, in practice, the choice of median vs. mean results in minor differences in the calibration outcomes (sec. A.3).

---

**Algorithm 1** Quantile calibration for pose estimation

**for** each $\mathbf{X}_m^*$ and $\mathbf{C}_m$ **do**
  draw $N$ hypotheses $\hat{\mathbf{X}} \mid \mathbf{C}_m$
  $\tilde{\mathbf{X}}_{m,k} \leftarrow \text{median}(\hat{\mathbf{X}}_{:,m,k})$
  $\varepsilon_{n,m,k} \leftarrow ||\hat{\mathbf{X}}_{n,m,k} - \tilde{\mathbf{X}}_{m,k}||_2$
  $\Phi_m(\varepsilon) \leftarrow \text{CDF}(\varepsilon_{:,m,k})$
  $\varepsilon_{m,k}^* \leftarrow ||\mathbf{X}_{m,k}^* - \tilde{\mathbf{X}}_{m,k}||_2$
**end for**
$\omega_k(q) \leftarrow \frac{1}{M} \sum_{m=1}^M \mathbf{1}_{\Phi_m(\varepsilon_{m,k}^*) \leq q}$
$\omega(q) \leftarrow \text{median}(\omega_k(q))$
$\text{ECE} = \frac{1}{|\mathcal{Q}|} \sum_{q \in \mathcal{Q}} |\omega(q) - q|$

---

NEW

FIX

### 3.2 SAMPLE-BASED METRICS PROMOTE MISCALIBRATION

In this section, we show that sample-based metrics are a major component that contributes to miscalibration. In principle, minMPJPE could be a good surrogate metric for NLL. However, as it

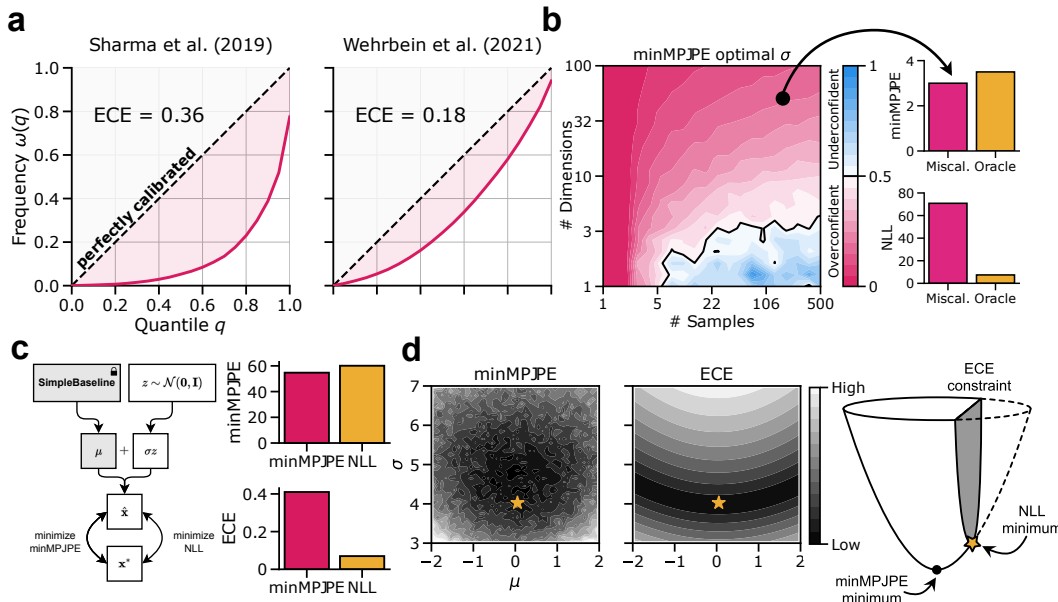

Figure 2: **a**) Calibration curves of previous lifting models with the corresponding expected calibration error (ECE) scores. **b**) Standard deviation $\sigma$ of a Gaussian distribution optimized to minimize minMPJPE for different numbers of samples and dimensions. The true $\sigma$ is 0.5 (black line), underconfident $\sigma > 0.5$ (blue), overconfident $\sigma < 0.5$ (pink). The human pose equivalent distribution (black point, 45 dimensions, 200 samples) compared to an oracle distribution (with true $\mu$ and $\sigma$) in terms of minMPJPE and NLL. **c**) Gaussian noise model schematic to the left. The SimpleBaseline model weights are not trained. Right bar plots compare the performance on minMPJPE and ECE when optimizing for minMPJPE and NLL. **d**) Loss landscapes of minMPJPE and ECE for a 1D Gaussian distribution with parameters $\sigma$ and $\mu$. The gold star represents the ground truth values of $\sigma^* = 4$ and $\mu^* = 0$. To the right is a schematic of the ECE constrained optimization.

became a common metric for selecting models it might become subject to Goodhart's Law (Goodhart, 1975) – "When a measure becomes a target, it ceases to be a good measure" (Strathern, 1997). In the case of minimizing the mean MPJPE over hypotheses, the posterior distribution collapses onto the mean (sec. A.1). Similarly, simulations indicate that minMPJPE converges to the correct mean, but it encourages miscalibration (Fig. 2b,d and sec. A.2).

We illustrate this with a small toy example. Consider $M$ samples $\boldsymbol{X}^* \in \mathbb{R}^{M \times D}$ from a $D$-dimensional Isotropic Normal distribution with mean $\boldsymbol{\mu}^* \in \mathbb{R}^D$ and variance $\boldsymbol{\sigma}^{*2} \in \mathbb{R}^D$ and an approximate isotropic Normal posterior distribution $q(\boldsymbol{X})$ with mean $\boldsymbol{\mu} \in \mathbb{R}^D$ and variance $\boldsymbol{\sigma}^2 \in \mathbb{R}^D$. We assume the ground truth mean to be known $\boldsymbol{\mu} = \boldsymbol{\mu}^*$ and only optimize the variance $\boldsymbol{\sigma}^2$ to minimize minMPJPE with $N$ hypotheses. We optimize $\boldsymbol{\sigma}^2$ for different numbers of dimensions $D$ and hypotheses $N$. If the distribution converges to a variance lower than the true variance $\boldsymbol{\sigma}^{*2}$ we call such a distribution **overconfident**. However, if the converged variance is larger than the true variance then the distribution is considered to be **underconfident**. Intuitively, for a small sampling budget drawing samples at the mean constitutes the least risk of generating a bad sample. With an increase in the number of hypotheses, increasing variance should gradually become beneficial, as the samples cover more of the volume. For a sufficiently large number of hypotheses, we can expect the variance to increase beyond the true variance, as the low-probability samples can have sufficient representation. Increasing dimensions should have an inverse effect since the volume to be covered increases with each dimension. We observe these effects in the toy example (Fig. 2b). When we consider the case which corresponds to the 3D pose estimation problem ($D = 45$ and $N = 200$, black point in Fig. 2b), we expect an overconfident distribution based on our toy example. This is also what we observe for the current state-of-the-art lifting models (Fig. 2a). Furthermore, we show that the minMPJPE optimal distribution outperforms the ground truth distribution in terms of minMPJPE, but not in terms of negative log-likelihood (Fig. 2b). Together, the results imply

NEW

that minimizing minMPJPE, directly or by model selection, is expected to result in miscalibrated distributions and thus minMPJPE by itself is not sufficient to identify the best model.

### 3.3 Unconditional Gaussian noise baseline on human 3.6m

To verify the conclusions from the toy model in section 3.2 we test the prediction with a simplified model on the Human3.6M dataset (Catalin Ionescu, 2011; Ionescu et al., 2014) (see section 5 for more details about the dataset). We train an additive Gaussian noise model on top of the SimpleBaseline (Martinez et al., 2017) a well-established single-hypothesis model. We generate $N$ hypotheses $\hat{\mathbf{X}} \in \mathbb{R}^{N \times M \times K \times 3}$ of poses with $K$ keypoints for $M$ observations $\mathbf{C} \in \mathbb{R}^{M \times K \times 2}$ according to:

$$\hat{\mathbf{X}}_{n,m} = \text{SimpleBaseline}(\mathbf{C}_m) + \boldsymbol{\sigma} \boldsymbol{z}_n$$

where $\text{SimpleBaseline}(\mathbf{C}_m)$ estimates the mean of the noise and $\boldsymbol{\sigma}$ is the standard deviation parameter scaling the standard normal samples $\boldsymbol{z} \sim \mathcal{N}(\boldsymbol{z}; \mathbf{0}, \boldsymbol{I})$ (Fig. 2c). It is important to note that we do not condition $\boldsymbol{\sigma}$ on the 2D observation $\mathbf{C}_m$, i.e. the same noise model is used for every input. We test two optimization setups: 1) minimizing minMPJPE and 2) maximizing likelihood. Based on the predictions from the toy problem (sec. 3), we expect the minMPJPE model to be overconfident and outperform the NLL model on the minMPJPE, but the NLL model to be better calibrated. This is exactly what we observe (Fig. 2c). Furthermore, each of these models achieve minMPJPE performance in a range similar to state-of-the-art multi-hypothesis methods and even outperform some established single-hypothesis methods (Table 1).  NEW

### 3.4 Evaluating sample-based methods

Given that minMPJPE is not sufficient to fully evaluate multi-hypothesis methods, we propose an evaluation strategy that remains interpretable and promotes calibrated distributions. Consider the landscapes of minMPJPE and ECE with respect to the mean and variance of an approximate distribution (Fig. 2d). Simulations indicate that optimizing minMPJPE identifies the correct mean  NEW
$\mu$ (sec. A.2), but not the correct $\sigma$. ECE, however, is minimized by a manifold of $\mu, \sigma$ values and converges to a good standard deviation for each mean, but it does not guarantee an accurate model.  NEW
We thus hypothesize that a likelihood-optimal distribution can be approximated when minMPJPE is minimized on the ECE-optimal manifold. Thus, minMPJPE can become a measure of *accuracy*  NEW
if it is constrained by ECE, but it should not be considered as accuracy if calibration is not matched.

## 4 Conditional graph normalizing flow

Given the observations made in sec. 3 we conclude that MPJPE-based objective functions are not  NEW
sufficient to obtain a well-calibrated distribution. The objective function should instead be based on likelihood, which in this case is maximized if and only if the estimated distribution recovers the ground truth distribution i.e. if the distribution is well-calibrated (Hastie et al., 2009). Therefore, in this section, we propose a method that can be optimized purely based on likelihood. Moreover, we utilize the natural graph structure of the human pose providing zero-shot generalization capabilities to occluded and unobserved body parts. We propose to learn the conditional distribution $p(\boldsymbol{x} \mid \boldsymbol{c})$ of the 3D pose $\boldsymbol{x}$ given the 2D pose $\boldsymbol{c}$ using conditional graph normalizing flows (cGNF).

We define a target graph $\boldsymbol{x} = (\boldsymbol{H}^x, \boldsymbol{E}^x)$ of 3D poses and a context graph $\boldsymbol{c} = (\boldsymbol{H}^c, \boldsymbol{E}^c)$ of 2D detections. $\boldsymbol{H}^x \in \mathbb{R}^{n \times D_x}$ and $\boldsymbol{E}^x$ are the edges between the nodes of the target graph and $\boldsymbol{H}^c \in \mathbb{R}^{m \times D_c}$ and $\boldsymbol{E}^c$ are the edges between the nodes of the context graph. In the case that an observation is not present, the corresponding node is removed from $\boldsymbol{c}$. The model is built of $L$ transformation blocks, each of which consists of a per-node feature split step, a graph merging step, an actnorm (Kingma & Dhariwal, 2018) and two graph neural network layers (Gori et al., 2005) (Fig. 3). These elements construct an affine coupling layer (Dinh et al., 2016), which is then followed by a permutation layer. The transformation blocks are only applied to the target graph, while the context graph is passed through unchanged.

**Per-Node Feature Split Step** splits the target node features $\boldsymbol{H}^x$ into two parts, $\boldsymbol{H}^x_{:,1:D-1}$ and  FIX
$\boldsymbol{H}^x_{:,D}$ across the feature dimension. We incorporate a leave-one-out strategy for splitting the features. The $i$th feature dimension is propagated directly to the affine coupling layer and the remaining

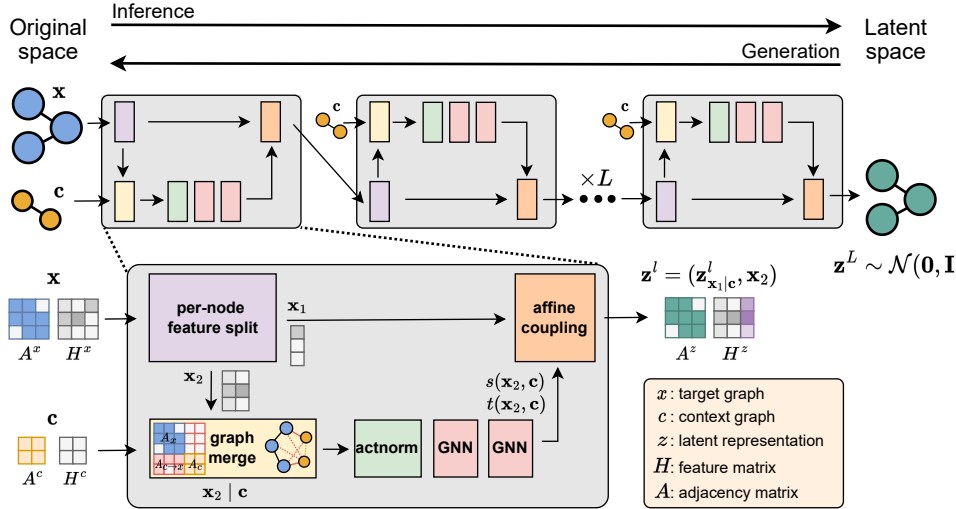

Figure 3: A schematic of the cGNF. Target variables $\boldsymbol{x}$ are represented by a graph with the feature matrix $H^x$ and the adjacency matrix $A^x$. The context variables are represented by a context graph $\boldsymbol{c}$ with the feature matrix $H^c$ and adjacency matrix $A^x$. In the inference path the target graph $\boldsymbol{x}$ is transformed into a latent space $\boldsymbol{z}$ which follows a standard normal distribution. The transformation is achieved through $L$ transformation blocks.

dimensions are passed to the graph neural network layers. In the next block, the next $i$th dimension is used.

**Graph Merging**   When utilizing conditional normalizing flows Winkler et al. (2019) on graph-structured data, a key challenge is incorporating the context graph in the transformation. We propose to merge the context graph $\boldsymbol{c}$ with the target graph $\boldsymbol{x}$ into a heterogeneous graph $\boldsymbol{x} \mid \boldsymbol{c}$. The context graph $\boldsymbol{c}$ forms directed edges from nodes in $\boldsymbol{c}$ to nodes in $\boldsymbol{x}$ as defined by $\boldsymbol{R}^{c \to x}$, the relations matrix. $\boldsymbol{R}_{i,j}^{c \to x} = 1$ indicates that node $i$ in the context graph forms an edge with node $j$ in the target graph $\boldsymbol{x}$ (Fig. 3).

**Graph Neural Network Layers**   We define the graph neural network layers as relational graph convolutions (R-GCNs) (Schlichtkrull et al., 2018). In the message passing step, the message received by node $v$ from the neighboring nodes is defined as

$$\mathbf{m}_{t+1}^{(v)} = \sum_{u \in \mathcal{N}^{c \to x}(v)} \psi_{c \to x}\left(\mathbf{h}_{\boldsymbol{c}}^{(v)}, e^{(u,v)}\right) + \sum_{r \in \mathcal{R}} \sum_{u \in \mathcal{N}^r(v)} \psi_r\left(\mathbf{h}_t^{(u)}, e^{(u,v)}\right)$$

where $\psi_r : \mathbb{R}^{D_n} \mapsto \mathbb{R}^{D_h}$ and $\psi_{c \to x} : \mathbb{R}^{D_c} \mapsto \mathbb{R}^{D_h}$, with $D_h$ as the number of latent dimensions. $\psi_{c \to x}$ should be flexible enough to allow the network to learn to distinguish between missing observations and zero observations i.e. $\psi_{c \to x}\left(\mathbf{0}, e^{(u,v)}\right) \neq \mathbf{0}$. The Update step is defined by the mapping $g : \mathbb{R}^{D_h} \mapsto \mathbb{R}^{D_o}$ which maps the latent space to the output dimension of size $D_o$. We implement the Update step as a single fully connected linear layer.

**Affine Coupling Layer**   Similarly to Liu et al. (2019) the output of the GNN layers models the scale $\mathbf{s}(\boldsymbol{x}_2, \boldsymbol{c})$ and translation $\mathbf{t}(\boldsymbol{x}_2, \boldsymbol{c})$ functions. The scale and translation functions are then applied to the unchanged split $\boldsymbol{x}_1$ to produce the transformed graph $\boldsymbol{z}_1^l$.

$$\boldsymbol{z}_1^l = \boldsymbol{x}_1 \odot \exp\left(\mathbf{s}(\boldsymbol{x}_2, \boldsymbol{c})\right) + \mathbf{t}(\boldsymbol{x}_2, \boldsymbol{c})$$
$$\boldsymbol{z}_2^l = \boldsymbol{x}_2$$

The $\boldsymbol{x}_2$ is copied to $\boldsymbol{z}_2^l$ unchanged. The $\boldsymbol{z}_1^l$ and $\boldsymbol{z}_2^l$ are then concatenated to form the transformed graph $\boldsymbol{z}^l$, which is passed to the next transformation block.

**Estimating Conditional and Marginal Densities**   The cGNF architecture allows for estimating   NEW
both the conditional and marginal densities within a single model. The conditional density $p(\boldsymbol{x} \mid \boldsymbol{c})$
is estimated by merging the target graph $\boldsymbol{x}$ with the context graph $\boldsymbol{c}$. Consequently, the output
density becomes constrained by the context. By removing nodes from $\boldsymbol{c}$, the associated conditioning
variables are removed from $p(\boldsymbol{x} \mid \boldsymbol{c})$, functionally conditioning only on a subset of the possible nodes
in $\boldsymbol{c}$. Finally, if the context graph is empty, the model provides a marginal density $p(\boldsymbol{x})$.

**Loss**   The standard optimization procedure for normalizing flows is to maximize the log probability
of the observed data $\boldsymbol{x}$ obtained through the inverse path $(\boldsymbol{x} \rightarrow \boldsymbol{z})$ (Fig. 3). Assuming $\boldsymbol{x}$ are i.i.d.
the task of the flow is to model $p(\boldsymbol{x} \mid \boldsymbol{c}) = \prod_i^N p(\boldsymbol{x}_i \mid \boldsymbol{c}_i)$ where $\boldsymbol{x}_i$ are the 3D poses and $\boldsymbol{c}_i$ are the
corresponding 2D observations. We thus define the loss as the negative log probability of pairs of
observations $\boldsymbol{x}$ and $\boldsymbol{c}$.

$$\mathcal{L}_{post.} = -\ln q_0(f(\boldsymbol{x}, \mathbf{c})) + \sum_{k=1}^{K} \ln \left| \det \nabla_{z_{k-1}} f_k(\boldsymbol{z}_{k-1}, \boldsymbol{c}) \right|$$

where $q_0 \sim \mathcal{N}(\boldsymbol{z}; \mathbf{0}, \boldsymbol{I})$ is the source distribution. We augment the training data by randomly re-
moving context variables to simulate new observations with missing keypoints in $\boldsymbol{c}$. The augmented
observations contain $20\%, 40\%, 60\%$ or $80\%$ of all observable keypoints. For all 3D poses, we addi-
tionally compute the prior loss, which expresses the likelihood of a pose given that no 2D keypoints
were observed.

$$\mathcal{L}_{prior} = -\ln q_0(f(\boldsymbol{x}, \varnothing)) + \sum_{k=1}^{K} \ln \left| \det \nabla_{z_{k-1}} f_k(\boldsymbol{z}_{k-1}, \varnothing) \right|$$

Our overall loss function is thus the sum of the two partial losses.   FIX

$$\mathcal{L} = \frac{1}{2} \Big( \mathcal{L}_{prior} + \mathcal{L}_{post} \Big) \tag{2}$$

The proposed training strategy and architecture formulate pose estimation as a zero-shot density
estimation problem. The cGNF model is trained on a subset of possible observations and is required
to evaluate previously unseen conditional densities. Such zero-shot capabilities are useful in reliably   NEW
estimating occluded poses. The graph structure allows the cGNF to share information between
nodes and as a result allows modeling distributions with sets of conditioning variables that have not
been seen before. We observe that cGNF can solve these zero-shot density estimation problems
comparably to specialized conditional normalizing flow problems (sec B.3).

**Root Node**   3D poses are relative to a root node (usually the pelvis). Hence, the root node's position
is deterministic. We, therefore, remove the root node and corresponding edges from the target graph
$\boldsymbol{x}$ and represent it as a *root* node-type $\mathbf{r}$, which has features $\boldsymbol{H}^{\mathbf{r}} \in \mathbb{R}^3$ and a message generation
function $\psi_r$ which is a fully connected neural network with 100 units.

**Graph Symmetries**   The human pose graph has symmetries, e.g. the left and right limbs are
mirrored. We impose a hierarchical structure on the nodes of the target graph $\boldsymbol{x}$. A node may have
a parent and a child, for example, the elbow node is the child of the shoulder node and the parent
of the wrist node. Messages passed from the parent to the child are *forward* messages generated by
$\psi_{x \rightarrow x}$ and messages from the child to the parent are *backward* messages generated by $\psi_{x \leftarrow x}$.

**Occlusion Representation**   We use 2D keypoint positions published by Wehrbein et al. (2021)
estimated using the HRNet model (Sun et al., 2019) and the provided Gaussian distribution fits for
evaluating occluded keypoints. If a keypoint is classified as occluded (2D detection $\sigma > 5$px) its
corresponding node is removed from the context graph. To adjust for the differences between the
pose definitions used by HRNet and H36M we employ an embedding network using the SageConv
architecture (Hamilton et al., 2017) with a learnable adjacency matrix. The embedding network
transforms the observed 2D keypoints into a 10-dimensional embedding vector for each of the key-
points. Additional implementation details of the architecture are given in the appendix (sec. B.1).

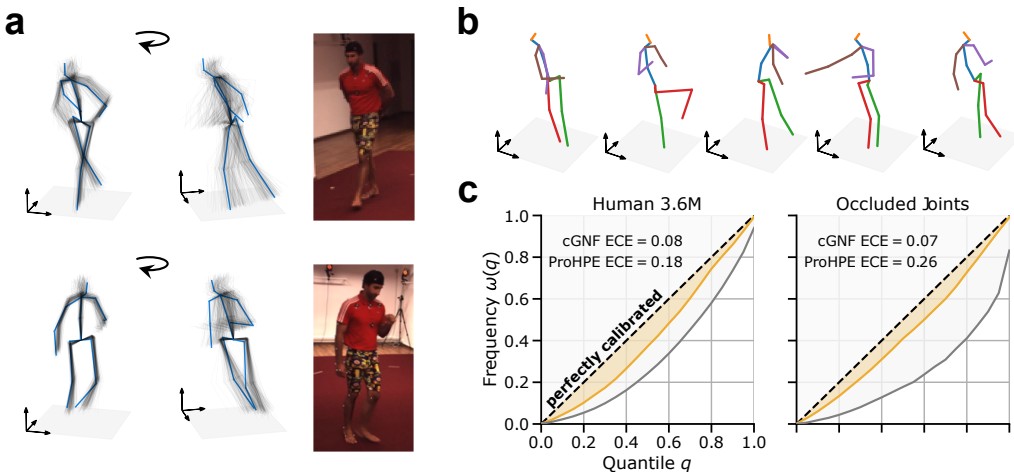

Figure 4: **a)** Hypotheses generated by the cGNF (gray) vs the ground truth pose (blue). Original image is shown to the right. **b)** Example of samples from the prior learned by the cGNF. **c)** Calibration of the conditional density. Comparison of the frequency that the distance of the ground truth from the median pose is within a given quantile. Median calibration curves for our model (cGNF, orange) and Wehrbein et al. (2021) (ProHPE, gray). Left shows calibration curves for the whole Human 3.6M test set and right for only the occluded joints.

## 5 LIFTING HUMAN3.6M

**Data**   We use the Human3.6M Dataset (H36M) on the academic use only license (Catalin Ionescu, 2011; Ionescu et al., 2014) which is the largest dataset for 3D human pose estimation. It consists of tuples of 2D images, 2D poses, and 3D poses for 7 professional actors performing 15 different activities captured with 4 cameras. Accurate 3D positions are obtained from 10 motion capture cameras and markers placed on the subjects. For evaluation, we additionally use the Human 3.6M Ambiguous (H36MA) dataset introduced by Wehrbein et al. (2021). H36MA is a subset of the H36M dataset containing only ambiguous poses from subjects 9 and 11. A pose is defined as ambiguous when the 2D keypoint detector is highly uncertain about at least one of the keypoints.

**Evaluation**   We evaluate the model on every 64th frame of subjects 9 and 11 and the H36MA subset. We compare our model's performance to prior work on $\mathrm{minMPJPE}$ and ECE using 200 samples (Table 1). As expected from the observations made in section 3, our method underperforms on $\mathrm{minMPJPE}$ but significantly outperforms on ECE (Fig. 4c). We further compare our method to NEW Kolotouros et al. (2021) which utilizes a similar likelihood-based loss and a normalizing flow architecture, but does not account for occlusions and does not utilize graph inductive biases (Table 2). As we predict in section 3 we find that Kolotouros et al. (2021) is well-calibrated. We show that cGNF outperforms Kolotouros et al. (2021) even though fewer samples are used and remains comparably well-calibrated. Samples from the posterior and prior are shown in figure 4a and b. Additional examples are included in the appendix (posterior samples Fig. S3; prior samples Fig. S4). We further NEW compare our model performance in scenarios where other models exhibit overconfidence (Fig. 1 and S6) and explore failure cases (Fig. S7).

**Performance on individual occluded joints**   The poses contained in H36MA are not only occluded but also generally more difficult than the average pose in H36M. Therefore, we propose to evaluate the performance on solely the occluded joints instead of the whole poses. We report these errors in table 1 (Occluded), where we show that our method outperforms the competing methods by a significant margin on both $\mathrm{minMPJPE}$ and ECE. Thus, this shows that our model is able to learn a posterior distribution that is more calibrated than previous methods and is able to outperform prior methods on $\mathrm{minMPJPE}$ for the occluded joints.

Table 1: Comparison of the cGNF model to state-of-the-art methods for multi-hypothesis pose estimation using expected calibration error (ECE) and minimum mean per joint position error (minMPJPE) between the ground truth 3D pose and $N$ hypotheses. Best model row is printed in bold font. Reporting the mean across the outcomes of 3 different seeds and the standard deviation (SD). For ECE the SD is smaller than 0.001 in all cases. Thus, we do not report the SD value for ECE in this table. For all the metrics lower is better. We underlined the results that we did not compute but instead used the originally reported value. With † we mark results which used ground truth 2D keypoints and not estimated 2D keypoints and these are not included in the comparison.

| Method | H36M (mm) | H36MA (mm) | ECE | Occluded (mm) | ECE | $N$ | # Params |
|---|---|---|---|---|---|---|---|
| Martinez et al. (2017) | 62.9 | - | - | - | - | 1 | 4,288,557 |
| Zhao et al. (2019) | 60.8 | - | - | - | - | 1 | 434,703 |
| Gaussian (minMPJPE) | 54.8 ± 0.002 | - | 0.42 (84%) | - | - | 200 | 4,288,572 |
| Gaussian (NLL) | 60.1 ± 0.002 | - | 0.07 (14%) | - | - | 200 | 4,288,572 |
| Kolotouros et al. (2021) (GT) | 37.1† | - | 0.07† (14%) | - | - | 200 | 25,475,744 |
| Li & Lee (2019) | 52.7 | 81.1 | - | - | - | 5 | 4,498,682 |
| Sharma et al. (2019) | 46.7 | 78.3 | 0.36 (72%) | - | - | 200 | 9,100,080 |
| Oikarinen et al. (2020) | 46.2 | - | 0.16 (32%) | - | - | 200 | 440,357 |
| Wehrbein et al. (2021) | **44.3** | **71.0** | 0.18 (36%) | 51.1 ± 0.13 | 0.26 (52%) | 200 | 2,157,176 |
| cGNF | 57.5 ± 0.06 | 87.3 ± 0.13 | **0.08** (16%) | 47.0 ± 0.18 | 0.07 (14%) | 200 | 852,546 |
| cGNF w $\mathcal{L}_{sample}$ | 53.0 ± 0.06 | 79.3 ± 0.05 | **0.08** (16%) | 41.8 ± 0.04 | **0.03** (6%) | 200 | 852,546 |
| cGNF xlarge w $\mathcal{L}_{sample}$ | 48.5 ± 0.02 | 72.6 ± 0.09 | 0.23 (46%) | **39.9** ± 0.05 | 0.07 (14%) | 200 | 8,318,741 |

**Improving minMPJPE performance and the effect on calibration** We can incorporate a couple of additional steps to improve the minMPJPE performance. We introduce an additional loss term

$$\mathcal{L}_{sample} = \mathrm{MPJPE}\left(\boldsymbol{x}^*, f^{-1}(\boldsymbol{0}, c)\right)$$

that encourages the model to predict the ground truth pose. The sample-based loss term is added to the vanilla loss (equation 2) with a scaling coefficient $\lambda_{sample} = 0.1$. Analogously to Kolotouros et al. (2021) we sample a pose from the mode of the source distribution and minimize the minMPJPE between the sampled pose and the ground truth pose. This additional loss term is shown to improve the minMPJPE performance. At the original model capacity, the minMPJPE and calibration performance show improvement. However, while the model performance on minMPJPE increases further with model capacity, calibration decreases significantly. We compare the performances in table 1. Additional model capacity evaluations are made in sec. B.4.

NEW

FIX

Table 2: Comparison of methods on the Procrustes-Aligned minMPJPE metric.

| Method | PA-MPJPE | $N$ |
|---|---|---|
| Kolotouros et al. (2021) | 42.4 mm | 4095 |
| cGNF w $\mathcal{L}_{sample}$ | **40.7 mm** | 200 |

## 6 CONCLUSION

In this study, we explored the problem of miscalibration in multi-hypothesis 3D pose estimation. Obtaining calibrated density estimates is important for safety-critical applications, such as healthcare or autonomous driving. Here we provide evidence that a focus on sample-based metrics for multi-hypothesis 3D pose estimation (e.g. minMPJPE) can lead to miscalibrated distributions. We propose a flexible model which can be trained to minimize the negative log-likelihood loss and show that, unlike previous methods, our model can learn a well-calibrated posterior distribution and outperforms comparably calibrated methods on minMPJPE. However, in particularly ambiguous situations, i.e. for the occluded joints, we show that our model outperforms the state-of-the-art on minMPJPE while maintaining a well-calibrated distribution. We believe that our findings will be useful for future work in identifying and mitigating miscalibration in multi-hypothesis pose estimation and will lead to more robust and safer applications of multi-hypothesis pose estimation.

NEW

REPRODUCIBILITY STATEMENT

Our code for reproducing the results is open-sourced at `https://github.com/XXXX`. We also include code to reproduce the results we obtained for other works. Experimental logs and downloadable pretrained models are fully available at `https://wandb.ai/XXXX`.    NEW

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

## A  METRICS

### A.1  MEAN PER JOINT POSITION ERROR

A popular optimization metric is the MPJPE. While this metric is especially popular in single-pose estimation methods, it has also been used in various forms in multi-hypothesis methods. Optimizing this metric causes the distribution of poses to be overconfident. We show this for a simple one-dimensional distribution, the generalization to the multi-dimensional case is straightforward. Given samples $x \sim p(x|c)$ from a data distribution given a particular context $c$, such as keypoints from a image, consider an approximate distribution $q(\hat{x}|c)$ supposed to reflect the uncertainty about $x|c$.

This below objective is equivalent to the mean position error for a single joint. Note that $x$ and $\hat{x}$ are conditionally independent given $c$, i.e. $x \perp \hat{x}|c$. The objective can then be expanded as follows:

$$
\begin{aligned}
\mathcal{L} &= \mathbb{E}_{x \sim p(x|c), \hat{x} \sim q(\hat{x}|c), c} \left[ (x - \hat{x})^2 \right] \\
&= \mathbb{E}_c \left[ \mathbb{E}_{x, \hat{x}|c} \left[ (x - \mu_c + \mu_c - \hat{x})^2 \right] \right] \\
&= \mathbb{E}_c \left[ \underbrace{\text{Var}[x \mid c]}_{\text{indep. of } q} - 2\mathbb{E}_{x, \hat{x}|c} \left[ (x - \mu_c)(\hat{x} - \mu_c) \right] + \mathbb{E}_{\hat{x}|c} \left[ (\hat{x} - \mu_c)^2 \right] \right] \\
&= \text{const.} - 2\mathbb{E}_c \left[ \underbrace{\mathbb{E}_{x|c} \left[ (x - \mu_c) \right]}_{=0} \mathbb{E}_{\hat{x}|c} \left[ (\hat{x} - \mu_c) \right] + \mathbb{E}_{\hat{x}|c} \left[ (\hat{x} - \mu_c)^2 \right] \right] \\
&= \text{const.} + \mathbb{E}_c \left[ \mathbb{E}_{\hat{x}|c} \left[ (\hat{x} - \mu_c)^2 \right] \right] \geq 0
\end{aligned}
$$

The expectation in the final line is non-negative and can be minimized by $q(\hat{x}|c) = \delta(\hat{x} - \mu_c)$, i.e. setting $\hat{x} = \mu_c$ and shrinking the variance to zero. This means that $q$ would be extremely overconfident.

### A.2 minMPJPE CONVERGES TO THE CORRECT MEAN

Consider 1D samples $x^*$ from a data distribution $p(x)$ and an approximate Gaussian distribution $q(x)$ with parameters $\mu$ and $\sigma$. We sample $N$ hypotheses from $q(x)$ and minimize the minMPJPE objective:

$$
\text{minMPJPE} = \mathbb{E}_{q(z)} \left[ \mathbb{E}_{p(x)} \left[ \min_i (x^* - \mu - \sigma z_i)^2 \right] \right]
$$

Consider $z_j^*$ as the $z_i$ sample which minimizes the expression for the $j$-th data sample $x_j^*$.

$$
\text{minMPJPE} = \mathbb{E}_{q(z)} \left[ \mathbb{E}_{p(x)} \left[ (x^* - \mu - \sigma z_j^*)^2 \right] \right]
$$

Thus the derivative can be computed to be

$$
\begin{aligned}
\frac{\partial}{\partial \mu} \text{minMPJPE} &= -2\mathbb{E}_{q(z)} \left[ \mathbb{E}_{p(x)} \left[ x^* - \mu - \sigma z_j^* \right] \right] = 0 \\
&= \mathbb{E}_{p(x)} \left[ x^* \right] - \mu - \mathbb{E}_{q(z)} \left[ z_j^* \right]
\end{aligned}
$$

Simulations indicate that $\mathbb{E}_{q(z)} \left[ z_j^* \right]$ can be approximated by a sigmoid function

$$
\mathbb{E}_{q(z)} \left[ z_j^* \right] = S \left( \mathbb{E}_{p(x)} \left[ x^* \right] - \mu \right) \cdot C(\sigma, N)
$$

where $C(\sigma, N)$ is a scalar scaling value dependent on $\sigma$ and the number of hypotheses. Thus the root of the derivative can be computed to be:

$$
\mu = \mathbb{E}_{p(x)} \left[ x^* \right]
$$

### A.3 IMPACT OF CENTER TENDENCY MEASURE ON EXPECTED CALIBRATION ERROR

The choice of center tendency measure should be considered when computing the expected calibration error. Therefore on a subset of the models presented in table 1 we compare the effect of choosing 3 different reference points. 1) The median of the samples 2) the mean of the samples and 3) the mode of the samples. We showcase the results in table 3. We observe that the use of median in contrast to mean has little to no effect on the computation of ECE. Using the mode as a reference point results in generally smaller values of ECE. Finally, it can be observed that regardless of the reference point type our cGNF model remains better calibrated than the other methods.

NEW

## B CONDITIONAL GRAPH NORMALIZING FLOW

### B.1 ARCHITECTURE DETAILS

The cGNF model consists of 10 flow layers. Each flow layer $f_k$ consists of two GNN layers each performing one message-passing step each as defined in eq. equation 4. In the first GNN layer each

Supplementary Table 3: Comparison of different reference points definitions on the resulting ECE score. In bold we mark the method that under the particular reference point has the lowest ECE.

| Method | Median | Mean | Mode |
|---|---|---|---|
| Sharma et al. (2019) | 0.36 | 0.36 | 0.14 |
| Wehrbein et al. (2021) | 0.18 | 0.18 | 0.08 |
| cGNF (ours) | **0.08** | **0.09** | **0.04** |

message generation function $\psi_r^{(1)}$ is a single layer fully-connected neural network with 100 units and a ReLU activation (Agarap, 2018). All the messages to a node are summed together resulting in the output of the MESSAGE as in eq. equation 4. Then the UPDATE step takes the message output as its input to a single-layer fully connected neural network with 100 units and linear activation. The context $c$ is transformed via $\psi_c$ to 100 dimensions and passed to the next GNN layer. In the next GNN layer, the message generation functions $\psi_r^{(2)}$ are single layer fully connected neural networks with 100 units and ReLU activation, The UPDATE is a neural network layer with 3 output units. In the next flow layer of the original context graph $c$ is used and not the transformed context.

## B.2 TRAINING DETAILS

We train the model on subjects 1, 5, 6, 7, and 8 on every 4th frame. We reduce the learning rate on plateau with an initial learning rate of 0.001 and patience of 10 steps reducing the learning rate by a factor of 10. Training is stopped after the 3rd decrease in the learning rate or 200 epochs. The model was trained on a single Nvidia Tesla V100 GPU, for about 6 days. FIX

## B.3 ZERO-SHOT DENSITY ESTIMATION

We evaluate cGNF's zero-shot capability to estimate a previously unseen conditional density. We simulated 50 different triple pendulums with initial velocities sampled from a normal distribution $v \sim \mathcal{N}(0, 10)$ for 25 timesteps each. Each pendulum was constructed from 4 nodes connected in a chain. The zeroth node was fixed and the remaining $x_1$, $x_2$ and $x_3$ were freely moving. The nodes were observed with $c_i = x_i + \varepsilon$ with $\varepsilon \sim \mathcal{N}(0, 5 \cdot 10^{-2})$. On this dataset, we trained 3 models. I. A CNF trained to estimate the density when all positions are observed $p(x \mid c_1, c_2, c_3)$ II. A CNF trained on a density where only one node is observed $p(x \mid c_1)$ III. A cGNF trained on the densities where at most 2 nodes are observed i.e. the cGNF never sees examples of $p(x \mid c_1, c_2, c_3)$.

To test zero-shot capabilities we compare the performances of these 3 models on $p(x \mid c_1, c_2, c_3)$. Model I (CNF) is used as reference for estimating this distribution when $p(x \mid c_1, c_2, c_3)$ is in distribution. Model II (CNF) is used to reference a model which cannot zero-shot estimate densities as it is out of distribution. Model III (cGNF) shows that our model can zero-shot estimate a previously unseen conditional density (Fig. S1).

## B.4 CONSEQUENCES OF MODEL SCALE

We explore the effect of increasing the number of parameters of the model. We train 3 sizes of models: 1) *small* with 852 546 parameters, 2) *large* with 3 301 546 parameters, and 3) *xlarge* with 8 318 741 parameters. The individual architectures were found by architecture search. We observe that as the size increases the performance of cGNF applied to the lifting task improves decreasing the gap to the state-of-the-art methods. The performance further improves outperforming the state-of-the-art method on occluded joints. However, the improvement in performance comes at a cost of calibration (Fig. S2).

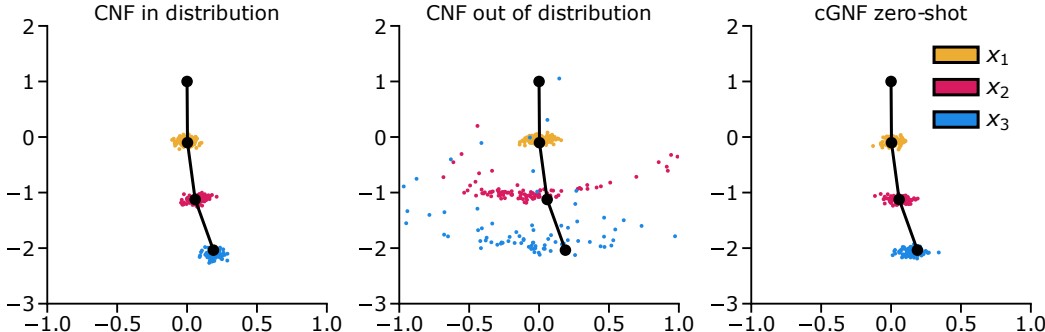

Supplementary Figure S1: Zero-shot capabilities of the cGNF model. Density estimates for 3 models: 1) CNF trained on $p(x|c_1, c_2, c_3)$ 2) CNF trained on $p(x|c_1)$ 3) cGNF which has never seen $p(x|c_1, c_2, c_3)$. The black points represent the true positions of the triple pendulum, and the orange, magenta and cyan represent samples for each node $x_1$, $x_2$, $x_3$ respectively.

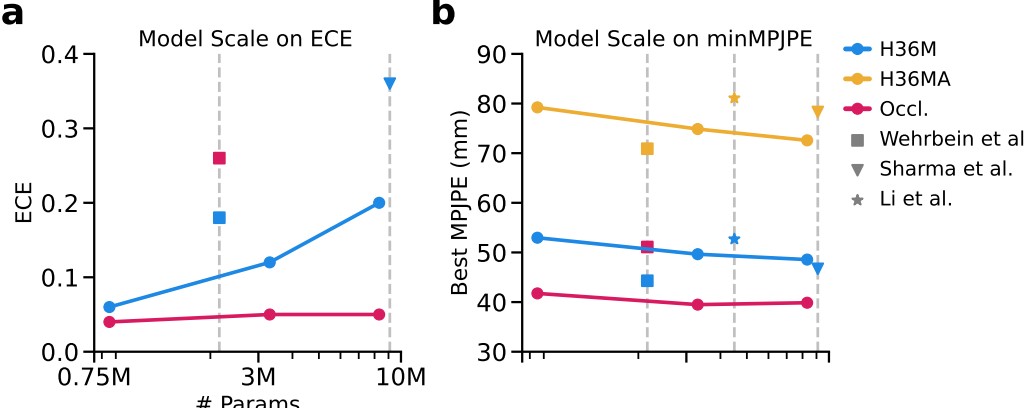

Supplementary Figure S2: minMPJPE and ECE across different model sizes. **a**) shows the effect of model scaling on calibration. **b**) shows the effect of model scale on accuracy. Performance on both metrics is compared to prior methods at their respective model sizes.

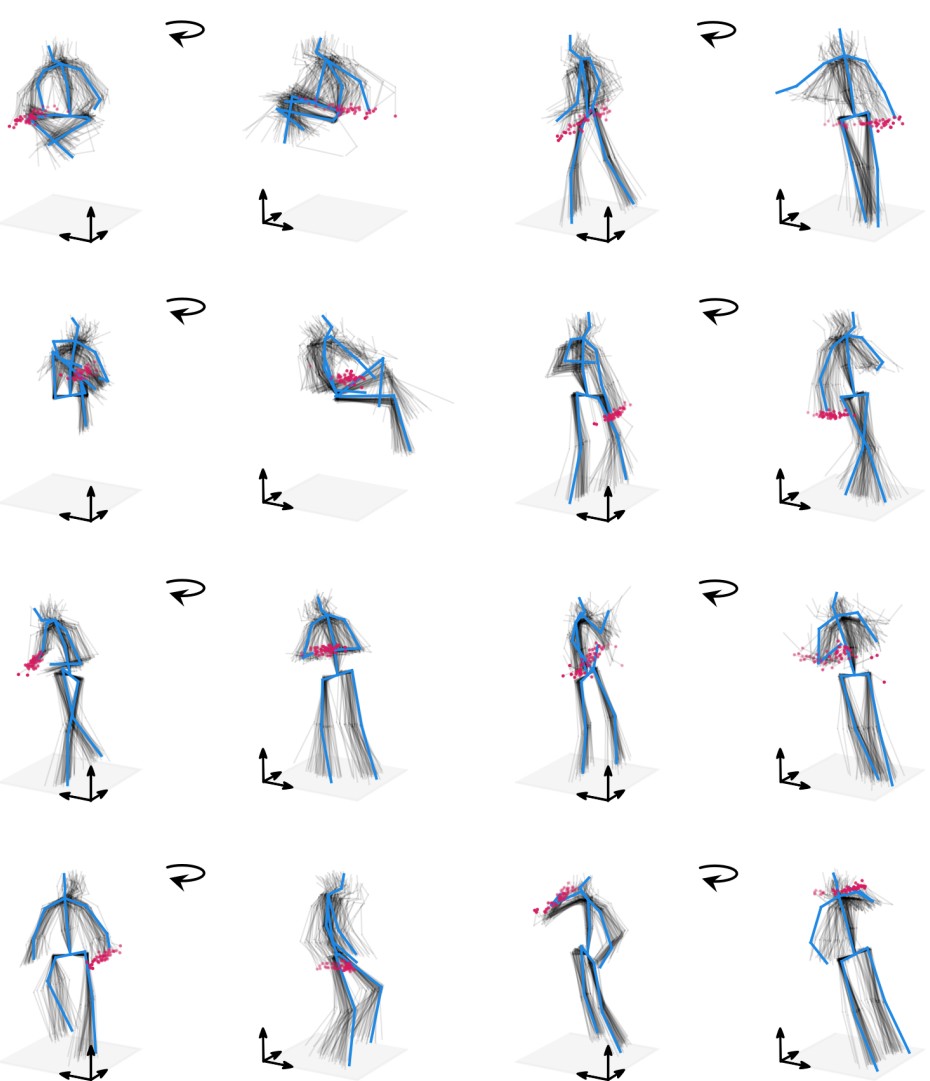

Supplementary Figure S3: Examples of samples from the posterior distribution learned by the cGNF (gray) vs the ground truth pose (blue). Pink points show the 50 sampled hypotheses for the right wrist positions. These examples are non-cherry picked and generated for subjects S9 and S11 from the test dataset.

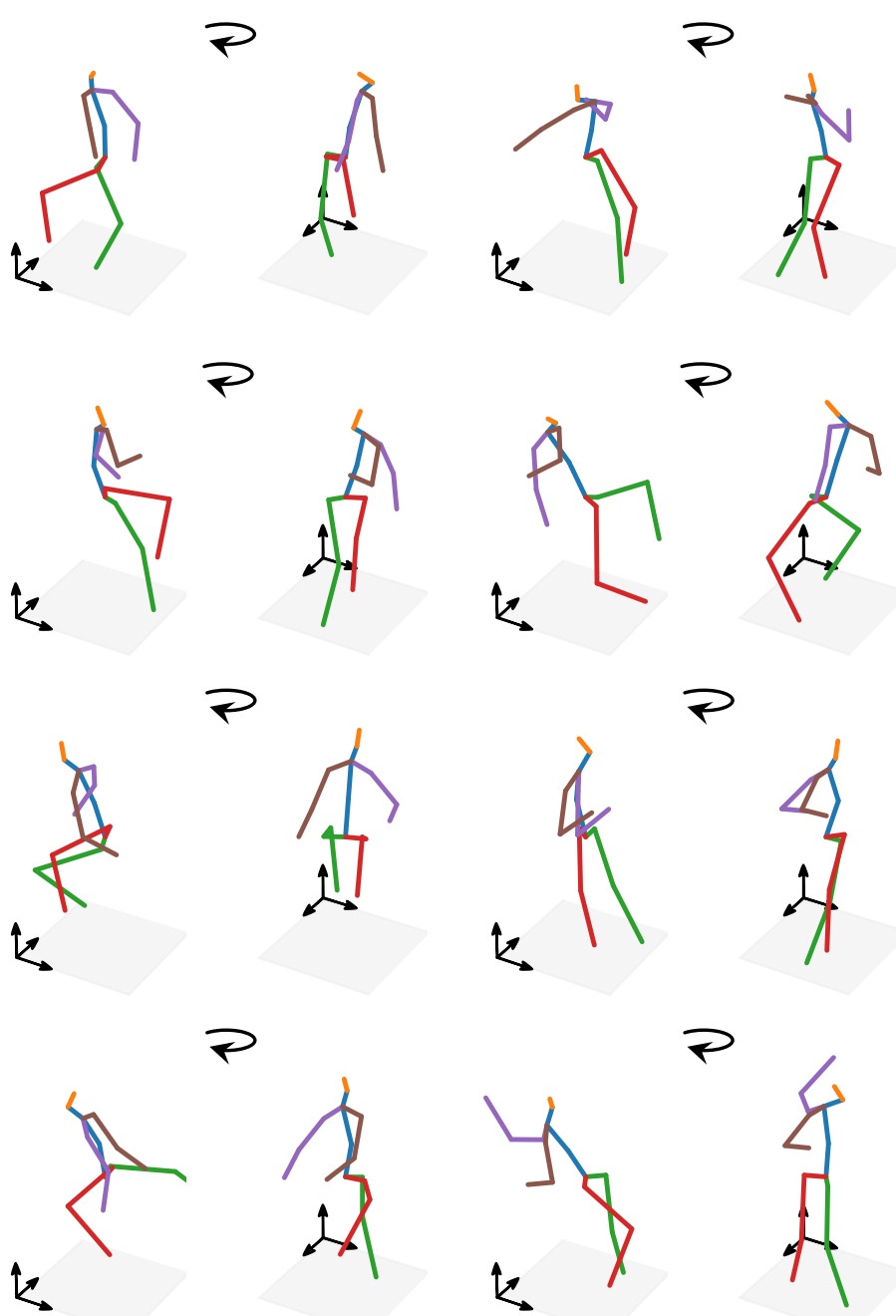

Supplementary Figure S4: Examples of non-cherry picked samples from the prior learned by the cGNF. Each are generated by randomly sampling a latent $z \sim \mathcal{N}(\mathbf{0}, \mathbf{I})$ and inverting to the pose space $\mathbf{x} = f^{-1}(z, \emptyset)$ without any context $\mathbf{c}$. For each pose images are shown for two rotations.

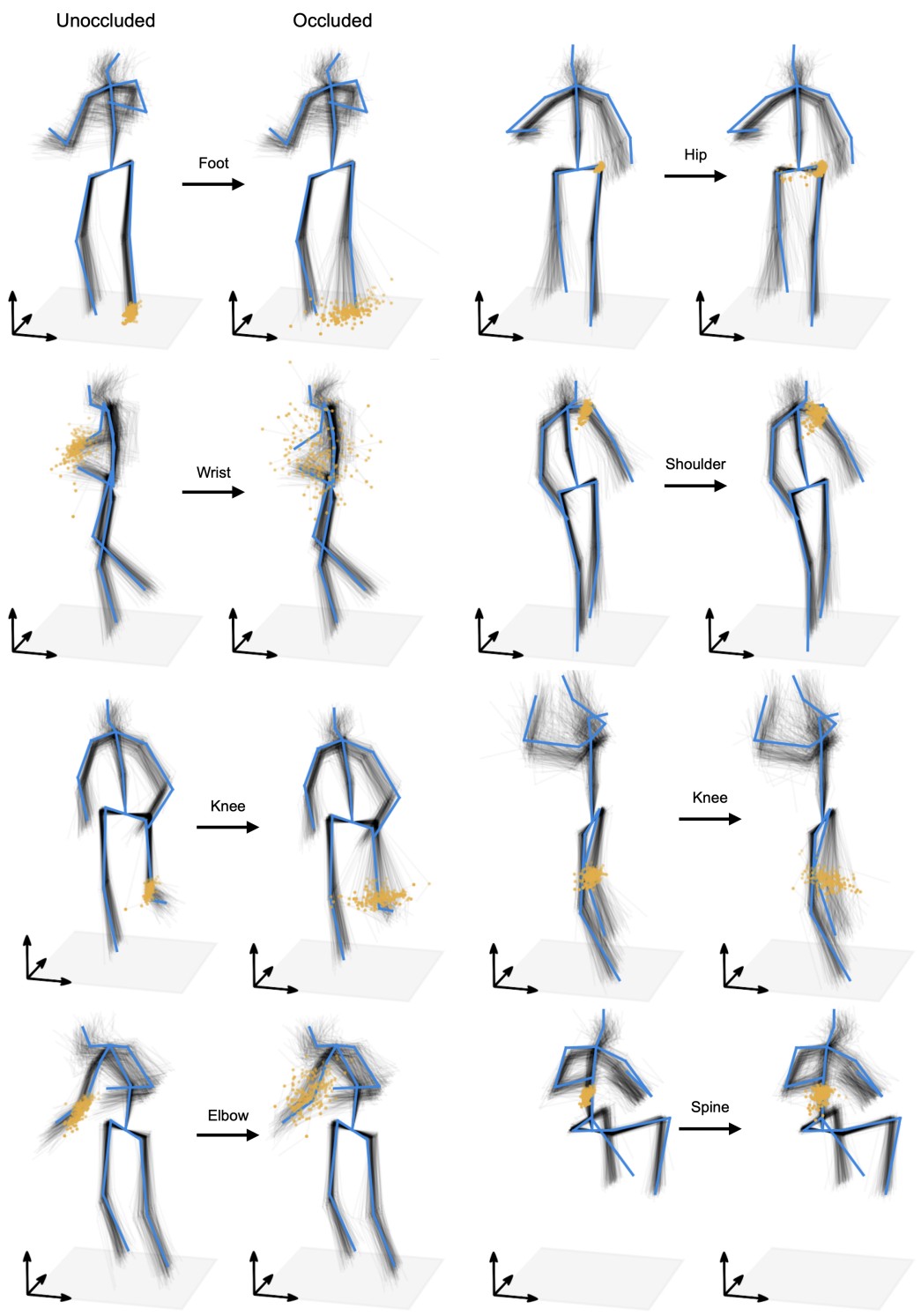

Supplementary Figure S5: Examples of the effect of artificially occluding a joint. Free joints like the foot or wrist show a high increase in variance, while internal joints like the shoulder and hip show smaller changes in variance. Gray poses mark samples from the cGNF and blue poses represent the ground truth 3D poses. Yellow points mark the sampled positions of the occluded joint.

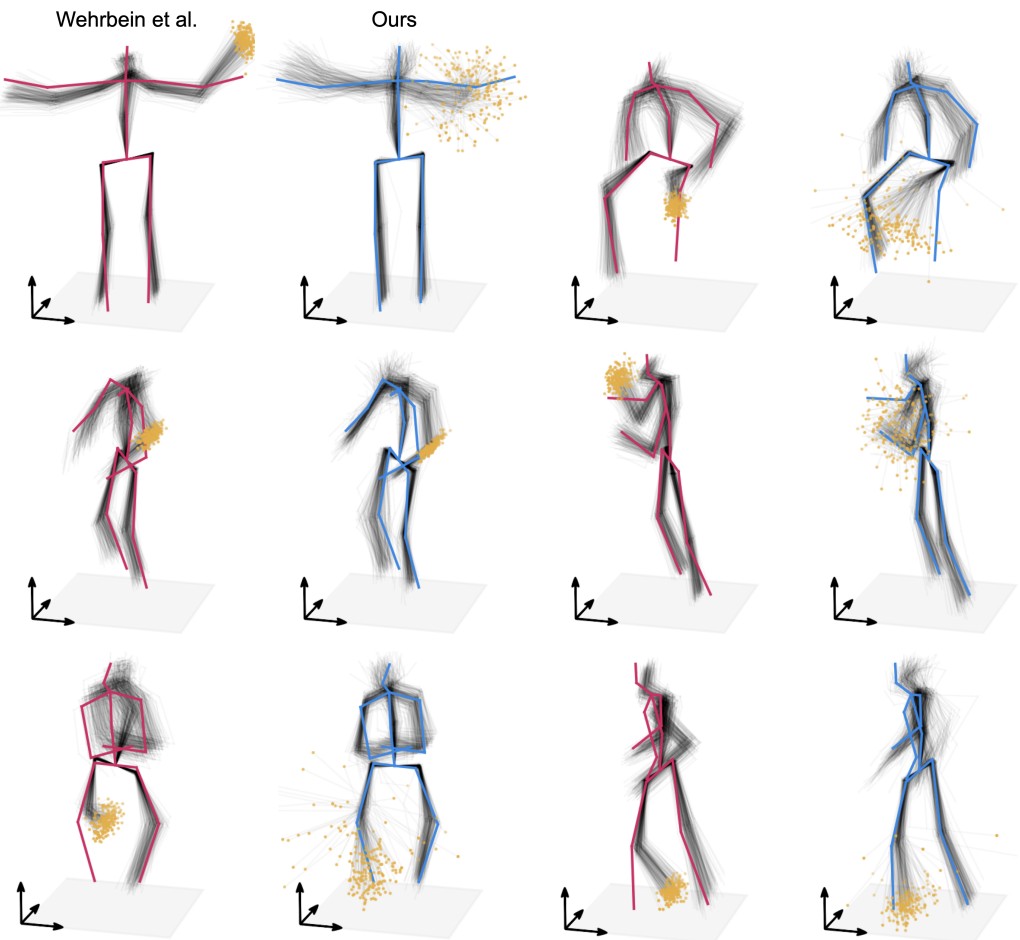

Supplementary Figure S6: Examples showcasing the consequences of overconfident distributions. Cases were artificially generated by shifting the 2d input and increasing the detection variance such that the joint is classified as occluded. Comparison between Wehrbein et al. failure cases where overconfident and wrong distributions were predicted and ours which produces a well-calibrated distribution. Gray poses mark samples from the cGNF and blue and pink poses represent the ground truth 3D poses. Yellow points mark the sampled positions of the joint of interest.

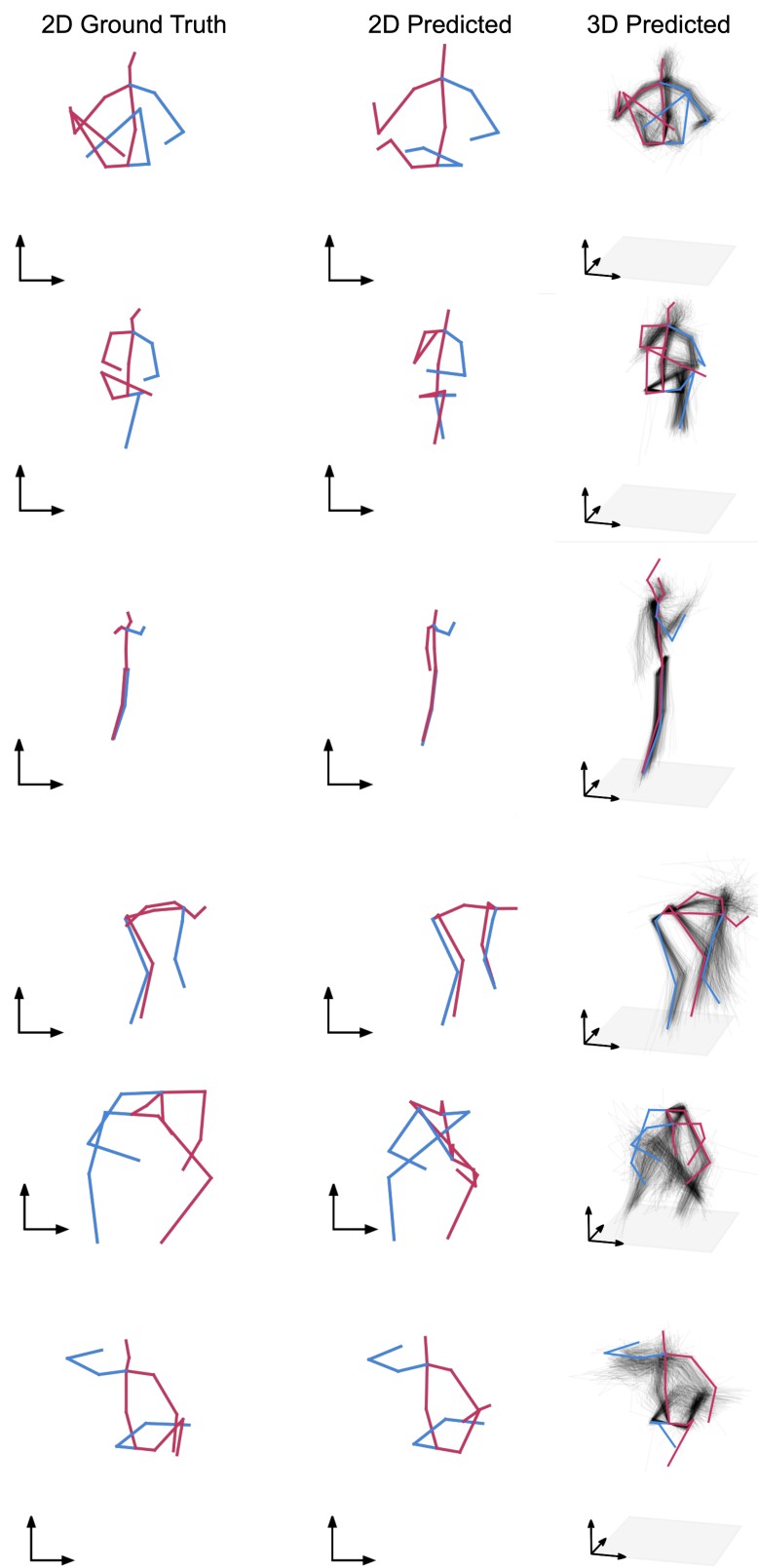

Supplementary Figure S7: Examples showing failure cases. For each of these poses, the minMPJPE exceeds 100 mm. In our experience, the most common cause of failure are over-confident and incorrect 2D detections. Gray poses mark samples from the cGNF and blue poses represent the ground truth 3D poses.

