# OpenReview forum: "Multi-Hypothesis 3D human pose estimation metrics favor miscalibrated distributions"
_ICLR.cc/2023/Conference — Submitted to ICLR 2023_

### Official Review · Reviewer_Sgm5 · 2022-10-24

**Confidence:** 3
**Correctness:** 3
**Technical Novelty And Significance:** 3
**Empirical Novelty And Significance:** 3
**Recommendation:** 3

**Clarity, Quality, Novelty And Reproducibility:**

Within the space limitations of the conference, I think the clarity of presentation is fine. The quality and novelty of this work are also without major issues, despite the fact that normalizing flows have been used in other problems previously. Finally, the manuscript is clear enough to fully enable reproducibility of the work to the best of my understanding.

**Strength And Weaknesses:**

This work is generally well presented, and the results are convincing.

One issue regarding the motivation of the work is the actual need for a well-calibrated method, especially if there is a trade-off with respect to accuracy: One may favor over-confident and more accurate models over (slightly) less accurate ones that are well-calibrated. It is perhaps useful to "know when you don't know", but arguably it is more useful to have more accurate estimations on average.

Another issue that may not be as important is the use of isotropic distributions: for the problem at hand, arguably the uncertainty of depth is possibly higher than that for the other two dimensions. This in turn may skew the estimation of the variance when assuming isotropic behavior.

**Summary Of The Paper:**

This work proposes a method to lift 2D human body keypoints to 3D by estimating the distribution of 3D joint locations. Special care is taken to ensure that, apart from the correct estimation of the mean, the estimated distributions are also well calibrated: It is argued and experimentally shown with toy examples that sample-based metrics for multi-hypothesis generation lead to overconfident distribution with variance lower than the true one. To mitigate this, a normalizing-flow based approach is proposed that takes advantage of the natural graph structure of the human kinematics. The resulting approach regresses the mode of the estimated distribution with small accuracy loss compared to current SoTA methods, while being better calibrated.

**Summary Of The Review:**

After the issues raised by the other reviewers and the response of the authors, I think this work is not yet ready for acceptance in ICLR. Although there is a motivation for the problem, it is not clear whether the proposed approach tackles it explicitly.

---

> ### Author Response · Authors · 2022-11-09
> **Authors' response to Reviewer Sgm5**
>
> We appreciate that you consider our work to be well-presented and results to be convincing. As far as we can tell your points are:
> - The need for well-calibrated distributions could be better motivated.
> - Isotropic distributions are insufficient to capture the nature of the problem.
>
> **Question about the use of isotropic distributions** Do you mean the use of isotropic distributions in the toy problems in sec. 3? If so we agree fully, that isotropic distributions are not sufficient to capture the full picture of the problem of human pose estimation. In the toy example in sec. 3.2 we do use an Isotropic Normal distribution, however, the simulated data is also isotropic. In sec. 3.3 the toy problem on Human 3.6M data we utilize a non-isotropic Gaussian. The $\sigma$ vector is of size equal to the number of joints (K) times the number of dimensions (3). If this was not clear in the text we will make it more explicit. In case we misunderstood your concern, please let us know.
>
> We are positive that we can address your other concerns in an updated version of the manuscript which we will provide soon. Please let us know if we forgot something.

---

> > ### Author Response · Authors · 2022-11-16
> > **Authors' response to Reviewer Sgm5 (Updated Manuscript)**
> >
> > **Motivation for well-calibrated distributions** see the general comment. As we discuss in sec. 3.4 $\operatorname{minMPJPE}$ is not directly a measure of accuracy as it can be changed by altering the variance only (which we show in sec. 3.3). This additionally opens the question of the role of multi-hypothesis methods. We identify 2 potential roles. 1) To provide the quantification of the ambiguity that single-hypothesis methods cannot or 2) find a sample that is as close to the ground truth as possible. If the goal is to achieve 2) then indeed minimizing $\operatorname{minMPJPE}$ would provide this; however, in practice, the ground truth is not available and thus knowing which sample is closest to the ground truth is not possible. Hence, a miscalibrated model would not provide any real-life benefit over a single-hypothesis method. We, therefore, argue that a good quantification of the uncertainty provides useful insight, which single-hypothesis methods cannot.

---

### Official Review · Reviewer_AasC · 2022-10-24

**Confidence:** 4
**Correctness:** 3
**Technical Novelty And Significance:** 3
**Empirical Novelty And Significance:** 3
**Recommendation:** 6

**Clarity, Quality, Novelty And Reproducibility:**

This paper is well-written and clear. I find the analysis of the minMPJPE metric novel and well-motivated, though I find it hard to make a connection between the proposed model and the analysis.

**Strength And Weaknesses:**

## Strength

- Analysis of calibration and the metric minMPJPE: this work conducts a thorough analysis of the issues with the popular minMPJPE metric. Namely, probabilistic methods trained to minimize MPJPE tend to collapse to the mean and not account for the uncertainty (overconfident). By approximating the uncertainty of the pose by using model prediction error and uncertainty of the samples by using deviation from the median, the analysis shows that directly utilizing MPJPE as a loss/metric will cause the methods to produce a distribution that has high calibration error.
    - The authors conducted a series of well-thought-out toy examples to showcase this phenomenon.
- The proposed cGNF model achieves comparable results against SOTA method in terms of MPJPE while also producing well-calibrated pose distribution that captures the uncertainty of the input 2D pose.
- This work conducts a comprehensive experimental analysis of state-of-the-art methods and showcased the strength of the proposed method. The paper is also clear and well-written.

## Weakness

- While the proposed shortcoming of MPJPE and ECE metric makes intuitive sense, I find the proposed method is quite disconnected from the main motivation. It is hard for me to find how the design choices made for cGNF relate to a better measure of the underlying distribution. The proposed “training using subset of observation” is close to a masking strategy and the training loss is a fairly standard NLL loss for normalizing flow in pose estimation [2]. As a result, I could not make a connection between the objective of obtaining a better-clibrated model and the actual proposed method (is the innovation in the architecture? I could not make a connection there either). While I find the analysis interesting and well-designed and quantifies a known pose estimation issue well (that SOTA methods often do not measure uncertainty well), the method does not seem to draw insight from it.
    - The claimed to estimate both conditional (which I see) and marginal (which I do not see) using the cGNF model needs to be further explained.
    - I also do not see how this is a “zero-shot density estimation problem” and how randomly using a subset to train can lead to this. (It could be me not understanding it and if the authors could further elaborate on this I could consider raising the score.).
- While the paper focuses on quantifying uncertainty and occlusion in pose estimation, few examples and results were actually shown showcasing the strength of the model. It would significantly strengthen the claim if **extensive** visual examples could be shown the benefit of the model (e.g. uncertain 2D keypoints actually correspond to the more spread-out hypothesis, and, equally important, that 2D keypoints with little ambiguity leads to a model that is closer to the mean).

## Question to authors

- During the analysis of the miscalibration behavior in using minMPJPE, the models’ samples’ deviation from the median is used to construct the error distribution. The difference between the ground truth and the median is then used to approximate the ground truth error.
    - I am not sure how the second part approximates the actual uncertainty in the ground truth samples. Each ground truth sample $m$ has a unique uncertainty associated with it. For instance, occluded poses lead to more significant errors. In equation (3) the summation term lumps all of them together and forms a distribution. This amounts to measuring the uncertainty at a per-joint level ($\epsilon_{m, k}$) and not per sample level. On the other hand, the uncertainty of the model is done at a per-sample model. I understand that it would be difficult to measure the uncertainty at a per-sample model for ground truth data, but the current model seems questionable.

[1] Wehrbein, T., Rudolph, M., Rosenhahn, B., & Wandt, B. (2021). Probabilistic Monocular 3D Human Pose Estimation with Normalizing Flows. *2021 IEEE/CVF International Conference on Computer Vision (ICCV)*, 11179-11188.

[2] Kolotouros, N., Pavlakos, G., Jayaraman, D., & Daniilidis, K. (2021). Probabilistic Modeling for Human Mesh Recovery. *2021 IEEE/CVF International Conference on Computer Vision (ICCV)*
, 11585-11594.

Small error: right above page 3 equation (3), should it be $\epsilon_{m, k}^*$?

**Summary Of The Paper:**

This work focuses on two main aspects of lifting 2D pose into 3D: 1) proving that the popular MPJPE metric is causing models to be overconfident and not accounting for uncertainty, and proposing a new metric in pose estimation that also measures calibration, ECE; 2) proposing a new method, cGNF conditioned Graph Normalizing Flows, that performs comparably in the MPJPE spectrum while outperforming other methods in ECE.

**Summary Of The Review:**

I find the analysis presented in the paper novel and a good contribution to the community, while the proposed model is less so. There are a few remaining questions that I would like to discuss the authors and am open to more clarifications.

---

> ### Author Response · Authors · 2022-11-09
> **Authors' response to Reviewer AasC**
>
> We appreciate that you find our paper to be clear and well-written and that you find our analysis of the state-of-the-art methods to be comprehensive. As far as we can tell your points are:
> - The connection between the cGNF and the analysis is not clear.
> - It is not apparent how marginal density estimation is possible by the cGNF model.
> - It is not clear how the problem we are solving is a zero-shot problem.
> - You asked for an extensive visual showcase of the cGNF uncertainty capturing capabilities.
> - You are unsure how our definition of ECE approximates the actual uncertainty of the ground truth samples.
>
> **Question about additional visual examples** We will provide additional visual examples showcasing 1) examples of overconfidence in other models and compare these scenarios to our model. 2) provide comparisons of uncertainties for occluded and unoccluded joints. Please let us know if that would provide sufficiently extensive visualizations in your opinion.
>
> **Small Error** If we understand correctly, you mean $\varepsilon_{:, k}^*$ should be $\varepsilon_{m, k}^*$? If that is the case, then no. We compute the frequency over all M ground truth poses, which is indicated by $:$. We will make that more explicit in the text.
>
> We are confident that we can address your other concerns in an update of the manuscript which we will provide soon. Please let us know if we forgot something.

---

> > ### Author Response · Authors · 2022-11-16
> > **Authors' response to Reviewer AasC (Updated Manuscript)**
> >
> > **Connection of analysis to model** See general comment. Indeed the proposed NLL is a standard loss term. Yet the majority of the work in the field tends to rely on sample-based objectives rather than NLL, even in normalizing flow models [1]. cGNF relies on the conclusion from the analysis that the focus should remain on NLL. Thus, the design of cGNF is NLL oriented. Furthermore, cGNF allows for estimating the 3D pose distribution under missing/occluded 2D keypoint detections. The cGNF model can estimate the posterior distribution of a 3D pose conditioned on only some of the joints. Previously, methods would handle  occlusion by setting the value of a joint to 0 or adding variance of the 2D detection as a feature, both of which do not truly reflect a missing data point (since the keypoint could be at 0 or have high variance but be not missing) We compare the latter in Fig. S6 showing that such an approach can fail  to correctly capture the scenario of missing or incorrect data.
> > - **Zero-shot density estimation** We define the problem of pose estimation as a zero-shot density estimation problem. We define each conditioning setup (e.g. right wrist is occluded) as a separate conditional density e.g. $p(x | c_{\text{\\\\ right wrist}})$. Thus every time a different set of joints is provided to condition the 3D pose a new density needs to be estimated. At training time there is only a subset of the possible conditional densities that can be observed, as there can be exponentially many configurations of joints missing. Thus at inference time the model likely needs to estimate a previously unseen conditional density. We show in sec. B.3. that cGNF is capable of estimating a previously unseen conditional density at a similar performance to a normalizing flow specialized in estimating the particular density. Such a representation allows us to 1) optimize an NLL-first objective and 2) define occlusions properly as conditional densities with some variables unobserved.
> > - **Marginal and Conditional Density Estimation** following from the zero-shot density estimation formulation, cGNFs generally estimate conditional densities. However, in the scenario where no joints are observed, the cGNF functionally estimates the marginal distribution. We show examples in Fig. 4b (previously Fig. 3b) and Fig. S4.
> >
> > **Additional Visualizations** We added additional visualization as per your suggestion. We show 1) more examples of posterior samples (Fig. S3) 2) visualizations of the effect of occluding a joint (Fig. S5) which results in larger variance 3) comparisons of settings where overconfident models fail and cGNF adds density on the ground truth position  (Fig. 1 and Fig. S6) and 4) failure case analysis showing that the one (in our experience very common)  cause of failure in cGNF are overconfident and wrong 2D keypoint detections (Fig. S7).
> >
> > **ECE computation clarification** The goal of calibration is to measure whether the spread of the estimated distribution is consistent with the spread of the ground truth samples. Thus we compute a distribution of the spread in the estimated distribution. If we would then test this spread against samples from the same distribution we would find that samples are in e.g. the 0.8 quantile 80% of the time, i.e. it would be well-calibrated with respect to itself. Thus we test whether the ground truth samples follow the same pattern, i.e. we test how frequently do ground truth points fall into given quantiles of the estimated distribution. Thus informing us about the consistency of the predicted quantiles with respect to the ground truth.
> > - **Full-graph evaluation vs per-joint** It is indeed nontrivial to compute calibration at a full graph (sample)  level, as poses do not lie in a Euclidean space. Consequently, computing L2 errors between samples would not be meaningful. This is also an issue in the general evaluation of pose estimation methods, which utilize per-joint metrics rather than per-sample metrics.
> >
> > [1] Wehrbein, T., Rudolph, M., Rosenhahn, B., & Wandt, B. (2021). Probabilistic Monocular 3D Human Pose Estimation with Normalizing Flows. 2021 IEEE/CVF International Conference on Computer Vision (ICCV), 11179-11188.

---

> > > ### Comment · Reviewer_AasC · 2022-11-19
> > > **Response to updated manuscript.**
> > >
> > > Thanks for the detailed answer and updates. My concerns about zero-shot density estimation and conditional estimation are now addressed.
> > >
> > > My remaining concerns are:
> > >
> > > - I still do not see how the ECE computation is correct. The ground truth samples are not really "ground truth" as there is no real sampling happening for ground truth. The **difference** between ground truth and the **estimated** median is used to construct a proxy "ground truth" error distribution (which does not really measure the uncertainty in the ground truth data. It measures the uncertainty of the trained model of the ground truth sample). I do not have a good answer to how to measure ground truth error, but what is being done in the paper seems counterintuitive.
> > >
> > > - As I understand [2] does optimize the NLL loss, so it is a direct comparison with the cGNF model. Table 1 only includes the result using GT 2D keypoint (could the comparison be done with estimated 2D keypoints?).
> > >
> > > - For the visual results: I appreciate the added result in figure S5, and it does showcase the model capability well. I am though looking for results on real images (as H36M does include challenging poses with severe occlusion).
> > >
> > > [2] Kolotouros, N., Pavlakos, G., Jayaraman, D., & Daniilidis, K. (2021). Probabilistic Modeling for Human Mesh Recovery. 2021 IEEE/CVF International Conference on Computer Vision (ICCV) , 11585-11594.

---

> > > > ### Author Response · Authors · 2022-11-20
> > > > **Additional Clarification**
> > > >
> > > > Thank you for your comment. We are glad to see that your concerns regarding zero-shot density estimation and conditional estimation have been addressed.
> > > >
> > > > Regarding your remaining concerns:
> > > > - **ECE Computation** You are correct in observing that ECE does not measure the uncertainty in the ground truth data. Exactly as you noted, ECE measures the uncertainty of the trained model relative to the ground truth sample. That is the intended measure and is standard in quantile calibration [1]. ECE in quantile calibration measures whether the trained model is consistent with the ground truth data. A well-calibrated distribution, would not predict the position of the ground truth to be, for example, always at the median. Instead, 20% of the time the ground truth sample would be contained in the 0.2 quantile, 50% in the 0.5 quantile, and so on. ECE measures whether the trained model has this property. Please note that the median for ECE is a measure of the trained model distribution's central tendency.
> > > > - **Difference to the median** To use the quantile regression setting, we need to reduce the error to a univariate distribution. We checked that different choices of the anchor point (median, mean, max) do not change our conclusions (see sec. A.3). This does make an isotropy assumption. However, if a distribution is calibrated, then it is also calibrated in the univariate space. A distribution could appear more calibrated than it actually is if it had a systematic error along rotations around the median (which would leave the norm of the difference invariant). However, when visually inspecting the samples (e.g. see figs S3, S5, S6) we haven’t found any evidence that this happens. Thus, we still think that the ECE is a reasonable measure for calibration in our submission.
> > > > - **Comparison of methods** That is correct, [2] optimizes the NLL loss and is a direct comparison. We show in table 2 that even though [2] uses more samples our method outperforms it on PA-MPJPE (compared on predicted 2D keypoints). Regarding reproducing results from [2] we would like to refer you to our response to reviewer F59S (both public and private; if you cannot see the private response, please let us know and we will post it again here). In summary: For reasons we explain in the response to F59S, we cannot fully reproduce their results. However, we did reproduce their scores for an easier problem of pose estimation using 2D ground truth keypoints (biased against us).
> > > > - **Visualizations** Figure S5 does show poses estimated from *real images* in the Human3.6M dataset. We believe figure S5 provides the reader with an intuitive insight as to what they should expect when a joint is set to be occluded. Is that what you are referring to, or are you asking to see the corresponding raw images for those poses?
> > > >
> > > > [1] Hao Song, Tom Diethe, Meelis Kull, and Peter Flach. Distribution calibration for regression. arXiv [stat.ML], May 2019.
> > > >
> > > > [2] Kolotouros, N., Pavlakos, G., Jayaraman, D., & Daniilidis, K. (2021). Probabilistic Modeling for Human Mesh Recovery. 2021 IEEE/CVF International Conference on Computer Vision (ICCV), 11585-11594.

---

### Official Review · Reviewer_8uoB · 2022-11-03

**Confidence:** 4
**Correctness:** 3
**Technical Novelty And Significance:** 2
**Empirical Novelty And Significance:** 2
**Recommendation:** 3

**Clarity, Quality, Novelty And Reproducibility:**

Writing is not so clear. Some notations are inconsistent and not explained well.

The quality and novelty are good but lack more quantitative results to prove, and more theoretical proofs of cGNF.

Code is not provided.


**Strength And Weaknesses:**

Pros

- The topic and the perspective of this paper are interesting.
- The analysis and the plots demonstrate the miscalibration issue to some extent.

Cons

- Except for the $minMPJPE$ metric, how about other factors, such as the model itself? For example, the distribution of the multiple hypotheses is depending on the design of the normalizing flow model, therefore, the miscalibration of the variance may also originate from the capacity of the model itself.
- The logic is not so clear. I'm agreed that one of the miscalibration reasons is using $minMPJPE$. But how could cGNF solve this and does it have a theoretical guarantee? And there are already some works using normalizing flow as mentioned in the paper, what's the difference between cGNF and [1][2]?
- Experiments are too few. The goal of multi-hypothesis is to solve the ambiguity in monocular 3D pose estimation by approaching the real distribution, which is expected to improve the ability to generalize to a new scene. However, all the experiments are conducted on H36M, which is a quite restricted and constrained scenario. Therefore, could the method also solve the miscalibration issue on the unseen 3DHP dataset (which is a common practice to evaluate the generalization ability of the model in this field)?
- The authors also mention that the uncertainty could not be corrected captured due to the miscalibration, but no experiments about the uncertainty are presented using the cGNF. Could cGNF provide correct uncertainty? Quantitative or quality results may be needed.
- Writing is not so clear and consistent, some notations are not clearly explained, and some even have typos. E.g. the $H_c$ in Fig. 2 caption, is actually $\bf{H}^{c}$ in the main text of Sec. 4, right? If wrong, please correct me.

[1] Nikos Kolotouros, Georgios Pavlakos, Dinesh Jayaraman, and Kostas Daniilidis. Probabilistic modeling for human mesh recovery. In ICCV, 2021
[2] Tom Wehrbein, Marco Rudolph, Bodo Rosenhahn, and Bastian Wandt. Probabilistic monocular 3d
human pose estimation with normalizing flows. In International Conference on Computer Vision
(ICCV), October 2021

**Summary Of The Paper:**

This paper raises an interesting problem, which is the miscalibrated distributions caused by using $minMPJPE$ to choose the best estimates in multi-hypothesis 3D pose estimation. To resolve the miscalibration, the paper uses the existing ECE metric proposed by Naeini et al., and proposes cGNF to learn the conditional distribution $p(x|c)$.

**Summary Of The Review:**

This paper is novel and interesting, but is not ready for publication at ICLR at this stage due to limited analysis and experiments.

---

> ### Author Response · Authors · 2022-11-09
> **Authors' response to Reviewer 8uoB**
>
> We appreciate that you consider our problem and perspective interesting. As far as we can tell your points are:
> - What other factors can influence calibration?
> - The link between the analysis and the model is not clear.
> - You asked for evaluating cGNF’s capabilities of capturing uncertainties.
> - You suggested evaluating cGNF’s performance on other datasets.
>
> **Factors influencing calibration** We are unsure what you are referring to, because we have discussed how model capacity affects calibration in sec. 5 and B.3. We find results that are in line with [1] which performs an in-depth study of the influence of architecture on calibration. If we misunderstood what you had in mind please let us know.
>
> **Question about the uncertainty evaluation of cGNF** We are not certain that we understand your question. We provide the initial experiments (sec. 3, fig 1) to show that ECE is a measure of how well the model captures uncertainty. We then compare ECE for our model and the literature (Table 1). What type of uncertainty experiments would you envision?
>
> **Additional Datasets** You requested the evaluation of the generalization capabilities of cGNF on other datasets. While we agree that it would be generally an interesting addition, the focus of the paper is on the problem of calibration in lifting. Thus we believe that additional datasets would deviate from the main theme of the paper, distract the reader and not be feasible in the given timeframe of the rebuttal.
>
> We are confident that we can address your other concerns about the writing style and will provide an updated version of the manuscript soon. Please let us know if we forgot something beyond that.
>
> [1] Chuan Guo, Geoff Pleiss, Yu Sun, and Kilian Q Weinberger. On calibration of modern neural networks. June 2017.

---

> > ### Author Response · Authors · 2022-11-16
> > **Authors' response to Reviewer 8uoB (Updated Manuscript)**
> >
> > **Other factors influencing calibration** See general comment
> >
> > **Connection between analysis and proposed model** See general comment. Regarding the theoretical guarantee. cGNF’s objective is to maximize likelihood. Maximal likelihood is obtained by recovering the ground truth distribution, which by definition is well-calibrated [1].
> >
> > **Typo** Thank you for pointing out the typo. We corrected it in the figure and in the manuscript. We also double-checked the text to ensure that all notations which do not adhere to the ICLR recommended notation from the textbook, Deep Learning [2], are clearly explained.
> >
> > **Missing code** We, unfortunately, forgot to release the code along with the initial submission and did not indicate our intent to publish the code. We have now released the anonymized code in zip and will release it publicly on Github if the paper is accepted. We additionally added a reproducibility statement section in the manuscript, where we will add a link to the code and pretrained models and experimental logs in the final version.
> >
> > [1] Trevor Hastie, Robert Tibshirani, and Jerome Friedman. The elements of statistical learning: data mining, inference and prediction. Springer, 2 edition, 2009
> > [2] Ian J. Goodfellow, Yoshua Bengio and Aaron Courville, Deep Learning, MIT Press, 2016.

---

> > > ### Comment · Reviewer_8uoB · 2022-11-21
> > > **Response to authors**
> > >
> > > Thanks for the detailed responses and the updated manuscript. Some of my concerns about the **factors influencing calibration** and
> > > **uncertainty evaluation** has been addressed.
> > >
> > > My main concern is **the contribution of the cGNF**.  The conclusion from the analysis in Section 3 shows that the core of the calibration lies in using NLL optimization instead of the minMPJPE. Although ProHMR [1] aims to tackle human mesh estimation, it is a general framework that can be applied in pose estimation alone, which already uses the normalizing flow models and optimizes NLL. As stated in Section 3 and Section 5 (**Evaluation**), ProHMR is well-calibrated. Although I appreciate the study of the toy examples in Section 3, the contribution of cGNF is largely reduced for three reasons: (1) Using normalizing flow-based framework and NLL objectives have been already explored in ProHMR, and proved to be well-calibrated in the experiments; (2) The design of cGNF still could not have a good trade-off between MPJPE and ECE metrics as shown in Table 1, where the loss of MPJPE is quite large when having small ECE (13.2mm on H36M, 16.3mm on H36MA, compared to Wehrbein et al. (2021)); (3) There is no ablation study of each module in cGNF to see the effect. Compared with the simple architecture of ProHMR which also uses normalizing flow, these additional designs do not bring gains, which also reduces the contribution of cGNF.
> > >
> > > Other remarks
> > > - **Uncertainty**. The visual results in Figure S5 are a good illustration of the uncertainty when a joint is occluded. Perhaps it is appreciated to paste the corresponding images at the side for reference as well.
> > >
> > > - I still find several typos in the updated revisions as well as notation inconsistency between the main text and captions, and notations in figures. For example, the feature matrix notations in the caption of Figure 3 are not bold as in the main text, but they are actually the same thing.
> > >
> > > - **Additional datasets**. What I am concerned about is that for real applications or downstream tasks, the conclusions from the experiments only on a simple dataset H36M maybe not work on other complex or natural-scene datasets such as 3DHP and PW3D, limiting the application of this method.
> > >
> > >
> > > [1] Kolotouros, N., Pavlakos, G., Jayaraman, D., & Daniilidis, K. (2021). Probabilistic Modeling for Human Mesh Recovery. 2021 IEEE/CVF International Conference on Computer Vision (ICCV), 11585-11594.
> > > [2] Tom Wehrbein, Marco Rudolph, Bodo Rosenhahn, and Bastian Wandt. Probabilistic monocular 3d human pose estimation with normalizing flows. In International Conference on Computer Vision (ICCV), October 2021.

---

> > > > ### Author Response · Authors · 2022-11-21
> > > > **Additional Clarification**
> > > >
> > > > Thank you for your comment. We are glad to see that your concerns regarding factors influencing calibration and uncertainty evaluation have been addressed. We are happy that you think Figure S5 is a good illustration of the uncertainty of occluded joints.
> > > >
> > > > Regarding your remaining concerns:
> > > > - **NLL Objective** You are correct that the NLL is not new and we do not claim that we are the first to use NLL as an objective. Our intention is to demonstrate that NLL-based methods lead to better-calibrated models, while sample-based do not, even though sample-based methods may have lower minMPJPE than NLL-based methods. To the best of our knowledge, the multi-hypothesis works on HPE did not explicitly consider calibration or the influence of the loss function used on it.
> > > > - **Comparison to Kolotouros et al.** Indeed Kolotouros et al. utilize a similar architecture. As we predict, Kolotouros et al. are well-calibrated. However, as we show in Table 2, our cGNF outperforms Kolotouros et al.
> > > > - **ECE-minMPJPE tradeoff** We agree that we would like to achieve good minMPJPE and good calibration. However, as we show in section 3, there is a trade-off between the two. We further show that the ground truth distribution would not minimize minMPJPE, but does minimize ECE. Wehrbein et al. has low minMPJPE but is not calibrated. Kolotouros et al. on the other hand is well-calibrated but has worse MPJPE performance. Our cGNF model is similarly calibrated to Kolotouros et al. but outperforms their method on PMPJPE (even when using more samples for Kolotouros), thus cGNF provides an improvement in the space of calibrated models. It is currently not clear whether it is possible to achieve the performance of Wehrbein et al. on minMPJPE and be well-calibrated. We need to leave that to future research. Here is a summary table of the results for comparison:
> > > >
> > > > |Method |PMPJPE (mm)|ECE|N samples |
> > > > |-|-|-|-|
> > > > |Wehrbein et al.|32.4|0.18|200|
> > > > |Kolotouros et al.|42.4|0.07|4095|
> > > > |cGNF w $L_{sample}$|40.7|0.08|200|
> > > >
> > > > - **Zero-shot capabilities** An additional property of the cGNF (and thus a contribution of this paper) is the zero-shot density estimation capability of the model. In B.3. we show that cGNF can estimate previously unseen conditional densities at a similar performance to specialized CNFs. This is an important capability of our cGNF model compared to a vanilla flow model for HPE. The zero-shot capability is then utilized in improving the model’s performance on occluded joints while remaining well-calibrated.
> > > > - **Substantial improvement on occluded joints** Furthermore, cGNF provides a substantial boost in the performance on occluded joints, which is not observed in the previous methods.
> > > > - **Visualization** We can add the corresponding images at the side of figure S5 in the final version (as we are not allowed to make changes anymore at this point).
> > > > - **Additional Datasets** We understand and share your concerns. One of the motivations behind this project was a lack of robustness of lifting methods when used in the real world or on clinical populations, and particularly an ability to know when the outputs can be trusted. A limitation of the field is the absence of larger, more representative 3D datasets.
> > > > However, an advantage of  3D lifting models is that they only utilize the 2D input keypoints, and so the difference between datasets relies primarily on the quality of the 2D keypoint detector. As long as the keypoint detector is capable of robustly detecting the 2D joint positions, the lifting model is indifferent to changes in the backgrounds, lighting conditions, or settings. Furthermore, when evaluating on the Human 3.6M dataset we test on 2 previously unseen humans (as is standard). So the model cannot overfit to a single human. Additionally, the theoretical analysis of calibration and miscalibration analysis is not dataset specific and thus will generalize to other datasets. Similarly, the zero-shot capabilities of cGNF are not dataset-specific.
> > > > Formatting - indeed we have missed adjusting the formatting to bold in the captions and will fix that. However, we were not able to find additional typos in the text using a spellchecker. If you could point them out, we are happy to fix them in the final submission (we cannot make further edits).
> > > > - **Ablation Study** This is a new concern, so we have not addressed it earlier. In the current manuscript, we do compare the significance of the L_sample loss (table 1). We also performed a more extensive ablation study at an earlier stage in the development process. We could include these results.
> > > >
> > > > |Method|H36M (mm) |H36MA (mm)|Occl.(mm)|
> > > > |-|-|-|-|
> > > > |w/o embedding|72.7|103.5|63.8|
> > > > |w/o symmetry|78.6|109.8 |63.5|
> > > > |w/o L_sample|57.5|87.3|47.0|
> > > > |w L_sample | 53.0 |79.3|41.8|
> > > >
> > > > We additionally add a copy of the private discussion with reviewer F59S about the Kolotouros results for your convenience.

---

### Official Review · Reviewer_F59S · 2022-11-03

**Confidence:** 3
**Correctness:** 3
**Technical Novelty And Significance:** 2
**Empirical Novelty And Significance:** 2
**Recommendation:** 3

**Clarity, Quality, Novelty And Reproducibility:**

Quality
* This paper lacks sufficient details on Sec.3, which lowers its quality.

Clarity
* There are some theoretical analyses that need to be enhanced. The whole motivation and experiment are clear.

Originality
* The novelty is that it questions the plausibility of one widely used metric for evaluating 3D pose models, which has not been seen in other works.


**Strength And Weaknesses:**

Strength
* This paper starts with good motivation and provides a toy example to provide a possible explanation about the miscalibration caused by minMPJPE.
* The performance of the proposed method is impressing on occluded samples by both MPJPE and ECE metrics.

Weaknesses
* The introduction in Sec.3 is hard to follow. I think this section is significant for helping construct a basic understanding of how ECE is used in 3D pose regression. But I fail to read it when I came to Sec.3.1. The authors should explain why using median is reasonable. Why not use other statistics, such as mean? If I understand correctly, this cumulative distribution function should give the probability of being less than a certain distance. However, I think the probability of the distance between the ground truth pose and the median is what we need. In Figure 1, why the model is underconfident if the optimal variance is larger than 0.5?

* The comparison with Biggs et al. (2020), Kolotouros et al. (2021), Oikarinen et al., (2020) should be added. Biggs et al. (2020) and Kolotouros et al. (2021) share a similar idea with this paper that simultaneously optimizes likelihood and joint distance. Oikarinen et al., (2020) use 2D pose as input and train with GNN. I would suggest the authors add more discussions with these papers to highlight the differences and explain the advantages of cGNFs. These paper also publicize their code with accessible pretrained models, it would be better if the authors provide experimental comparisons.

* Although the good performance could be seen on ECE metric, the loss on MPJPE metric is also noticeable. Through the measurement of the article design, it seems that this method has made a great contribution to improving the credibility of the model. However, the results also show that certain gains on ECE would result in a loss on MPJPE. It’s hard to say whether such consequences are acceptable to us, especially when the loss is not subtle (7.7mm increase on MPJPE comparing cGNF w L_sample and Wehrbein et al. (2021)).

[1] Biggs, Benjamin, et al. 3D Multi-bodies: Fitting Sets of Plausible 3D Human Models to Ambiguous Image Data, 2020

[2] Nikos Kolotouros, Georgios Pavlakos, Dinesh Jayaraman, and Kostas Daniilidis. Probabilistic modeling
for human mesh recovery, 2021

[3] Tuomas P. Oikarinen, Daniel C. Hannah, and Sohrob Kazerounian. Graphmdn: Leveraging graph
structure and deep learning to solve inverse problems, 2020

**Summary Of The Paper:**

This paper reveals an interesting finding that minMPJPE metric would result in miscalibrated distribution in a 2D-to-3D lifting problem, which is particularly unsuitable for using MPJPE to evaluate those distribution-based methods that generate multiple hypotheses to solve the ambiguity. It provides a toy example that shows optimizing minMPJPE would have a larger ECE than NLL and lead to overconfident distribution, while a good NLL may not guarantee excellent MPJPE, but helps to calibrate the distribution. To this end, they propose cGNFs and train it by likelihood maximization to learn the conditional 3D human pose distribution. Together with optimizing minMPJPE, their method would achieve comparable performance to SOTA. From experimental results, their method has better ECE performance on H36M, and an obvious boost in MPJPE can be seen on occluded samples.

**Summary Of The Review:**

This paper starts with a nice motivation, but the writing and presentation are suboptimal. This paper claims the necessity of using ECE to evaluate the regression model, but I think Sec.3 is not solid enough to let me believe the way they formulate ECE is correct, which further affects the credibility of comparison results in Sec.5. I would consider changing my rating If the above comments are addressed.

---

> ### Author Response · Authors · 2022-11-09
> **Authors' response to Reviewer F59S**
>
> We appreciate that you consider our finding interesting. As far as we can tell your points are:
> - The section introducing ECE for pose estimation was not clear.
> - cGNF should be compared to Biggs et al. (2020), Kolotouros et al. (2021), Oikarinen et al., (2020)
> - You are concerned about the performance decrease in minMPJPE.
>
> **Comparison to additional methods** We shall compare to these methods. However, Biggs et al. (2020) although a normalizing flow model, perform the task of human mesh recovery from images instead of lifting 2D keypoints to 3D. Therefore our model and the models explored in this paper would not be directly comparable to the methods proposed by Biggs et al.
>
> We are positive that we can address your concerns in an update of the manuscript which we will provide soon, please let us know if we forgot something.

---

> > ### Author Response · Authors · 2022-11-16
> > **Authors' response to Reviewer F59S (Updated Manuscript)**
> >
> > **Expected calibration error section**
> >  - **Clarity** We rewrote section 3.1 to improve clarity.
> >  - **Choice of median** We use the median as it is robust to outlier samples. We compare different methods for choosing the reference point in sec. A.3. We show that there is in practice little difference between the choice of mean and median. We additionally compare to the mode as a reference point. In that case, all model ECEs are smaller. However, regardless of the choice of reference point cGNF is better calibrated than other methods.
> > - **Probability of distance** In general, we agree that the measure we would want to obtain is the probability of the ground truth. However, quantile calibration provides a proxy for the probability, which for many models is not available. It measures the frequency of the ground truth point falling into each quantile. Thus it doesn’t directly measure the probability, but the consistency of the spread.
> > - **Over- and underconfidence definition** We added a clearer definition of over and underconfidence in the manuscript. We define a distribution to be overconfident if its variance is lower than the true distribution’s variance as it assumes less uncertainty than there actually is. An underconfident distribution is one where the estimated variance is larger than the true variance, i.e. it assumes more uncertainty than it needs to. Thus, in Fig. 2 (previously Fig. 1) when the $\operatorname{minMPJPE}$-optimal variance is larger than the true variance (0.5) then the model is underconfident.
> >
> > **Comparison to other methods**
> > We performed additional comparisons to the suggested methods.
> >
> > | Method | minMPJPE | ECE | # Samples |
> > | - | - | - | - |
> > | Kolotouros et al. (2D Ground Truth) | 32.9 | - | 4096 |
> > | Kolotouros et al. (2D Ground Truth) | 37.1 | 0.07 |  200 |
> > | Oikarinen et al. (2D Ground Truth) | 31.8 | - | 200 |
> > | Oikarinen et al. (Predicted 2D) | 46.2 | 0.16 | 200 |
> > | cGNF w L_sample (Predicted 2D) | 53.0 | 0.08 | 200 |
> >
> > We additionally compare the performances on Procrustes Aligned minMPJPE.
> >
> > | Method | PA-minMPJPE | # Samples |
> > |-|-|-|
> > | Kolotouros et al. (Predicted 2D) | 42.4 | 4095 |
> > | cGNF w Lsample | 40.7 | 200 |
> >
> > Given that we outperform Kolotouros et al. with substantially fewer samples and are comparably calibrated, we believe that this shows that the architecture of cGNF is capable of outperforming Kolotouros et al. while remaining well-calibrated.
> >
> > **Loss in minMPJPE performance** As we discuss in sec. 3.4 $\operatorname{minMPJPE}$ without the knowledge of calibration is not a direct measure of accuracy: As we show in Fig. 2d (previously Fig. 1d) $\operatorname{minMPJPE}$ can be improved by changing the mean (accuracy) *and* the variance. Thus if variance is not constrained the differences in $\operatorname{minMPJPE}$ do not necessarily reflect changes in accuracy (mean). As we show in sec. 3.3 even in the simple gaussian model just changing the variance of a distribution can provide a 5.3 mm decrease in $\operatorname{minMPJPE}$. In the light that the SOTA models in $\operatorname{minMPJPE}$ are miscalibrated, they are not directly comparable in terms of accuracy. However, a comparison is possible when calibration is matched: In the comparison between Kolotouros et al. and cGNF which show similar calibration (0.07 and 0.08), the difference in $\operatorname{minMPJPE}$ is indicative of the difference in accuracy.

---

> > > ### Comment · Reviewer_F59S · 2022-12-11
> > > **Response to authors**
> > >
> > > Thank you for the update. It solves some questions but there are still many concerns and the paper is not ready for publication.  I list some major issues below:
> > > * **Soundness of the proposed ECE metric.** The two heuristic median operations are included in the definition and calculation of the proposed ECE metric, making it seem less sound than the version for classification. In the ICLR’21 paper[1], they studied a unified calibration metric for arbitrary dimensions and claimed that the quantile-based calibration cannot correctly reflect the distribution. The argument in [1] seems more plausible and contradicts the formulation of ECE in this work.
> > > * **The zero-shot (density estimation) formulation of HPE with occlusion is unclear.** It is quite different from the conventional DL in-distribution generalization formulation [2]. This paper doesn’t follow other related works but also does not give sufficient justification for this zero-shot setting. Most performance boosts of this work can be seen in the occluded scenarios (Table 1), and zero-shot is a reason for introducing GNFs.
> > > * **The unavoidable trade-off between minMPJPE and calibration.** It’s nice to see the added comparisons with Oikarinen et al. and Kolotouros et al.  but concerns remain regarding the trade-off between minMPJPE and calibration. The significant loss of 8.7mm for minMPJPE (cGNF w L_sample vs. Wehrbein et al. (2021)) on H36M cannot be ignored and begs the question if the 0.1 gain on calibration is worth it.  This depends on the plausibility of ECE and goes back to the concern on the soundness of the metric.
> > > * **The problem in theorem proof.** The last 3 rows of equation derivation in Sec. A.2 has problems. It cannot be obtained that the root of the derivate at optima, ∂minMPJPE/∂μ = 0, is the correct mean, since 0 - Sigmoid(0) * C(σ, N) ≠ 0. It seems that the minMPJPE objective has many local optima (μ, σ), and the correct GT mean is just one of them  (also see Fig. 2d col. 1). So optimizing minMPJPE cannot be guaranteed to converge to the correct mean.
> > >
> > > The idea of introducing uncertainty measures to HPE is good and valuable to explore in the community. However, the above concerns must be resolved in a convincing way.  The writing also needs to be adjusted to highlight the main contributions. I encourage the authors to consider the reviews and improve the work for future submission.
> > >
> > > [1] Widmann, David, Fredrik Lindsten, and Dave Zachariah. "Calibration tests beyond classification." ICLR, 2021.
> > >
> > > [2] Goodfellow, Ian, Yoshua Bengio, and Aaron Courville. Deep learning. MIT press, 2016.

---

### Author Response · Authors · 2022-11-09
**Authors' General Response**

We thank the reviewers for their helpful and constructive feedback. We were happy to see that the reviewers found our manuscript to be “well written” (Rev **AasC**), “well presented” (Rev **Sgm5**), and our work to be “novel” (Rev **8uoB**, **AasC**, **Sgm5**), “interesting” (Rev **F59S**, **8uoB**, **AasC**) and “not been seen in other works” (Rev **F59S**) remarking that “The analysis is a good contribution to the community” (Rev **AasC**). However, they also raised a few points of criticism, which we would summarize into the following key points:
- Clarify the definition of the expected calibration error (Rev **F59S**, **AasC**)
- Clarify the significance of well-calibrated models and the tradeoff between minMPJPE and ECE. (Rev **Sgm5**, **F59S**)
- Clarify the connection between the analysis and the proposed model (Rev **Sgm5**, **8uoB**)
- Clarify which other factors can influence calibration (Rev **8uoB**)
- Compare the calibration of additional methods (Rev **F59S**)
- Additional visualizations of the uncertainties (Rev **AasC**)

We are confident that we can address all of the above and other minor concerns.

In addition, Rev **8uoB** asked us to explore the generalization capabilities of cGNFs to other datasets. Because we want to focus on the problem of calibration in liftings we don’t think we can provide meaningful insights on other datasets in the given timeframe and will thus not pursue this request further in the rebuttal.

**Code Release** We will release the code for this paper allowing full reproduction of the results. We unfortunately forgot to mention that in the original submission or to submit the code as zip. We now attached a zip file with the code anonymized and if the paper is accepted we will release it on Github.

We will already reply to some of the minor issues now, and provide detailed answers to questions that require additional experiments in a few days.

---

> ### Author Response · Authors · 2022-11-16
> **Authors' General Response (Updated Manuscript)**
>
> **Clarity of ECE definition** We rewrote section 3.1 to improve clarity. We added explanations to the individual questions in separate comments to each of the reviewers.
>
> ​​**Significance of well-calibrated models** We added additional discussion of the conclusions in section 3 (Observing Miscalibration). We consider 2 roles for multi-hypothesis models. 1) To provide the quantification of the ambiguity that single-hypothesis methods cannot or 2) find a sample that is as close to the ground truth as possible. To achieve 1) we need a well-calibrated distribution. By minimizing $\operatorname{minMPJPE}$ we can achieve 2). However, when using these models, the ground truth is not available to select the best sample and we argue this metric alone has no practical meaning and does not reflect any real accuracy. A miscalibrated multi-hypothesis model provides no additional benefit over single-hypothesis methods. As we show in sec. 3.3 a well-calibrated model even though it has higher $\operatorname{minMPJPE}$ is not necessarily less accurate (the mean of the miscalibrated and calibrated models are the same).
> Therefore, we argue that $\operatorname{minMPJPE}$ by itself is not a meaningful measure of accuracy since changing the variance can change the $\operatorname{minMPJPE}$ metric without increasing accuracy. We, therefore, indicate that the observed decrease in $\operatorname{minMPJPE}$ for cGNF is not directly indicative of a loss in accuracy. We additionally provide visualizations of cases where overconfident models fail to capture the ground truth pose (and are very confidently wrong) while a well-calibrated model places probability density in the position of the ground truth pose and provides higher variance of samples.
>
> **Analysis and proposed model connection** We added a new paragraph at the beginning of section 4. Our conclusion from the miscalibration analysis is that maximizing likelihood should result in a well-calibrated model. We, therefore, designed a method that can optimize an objective that is purely likelihood-based. We additionally employ a graph-based representation as it provides a natural way to represent the human pose and allows us to naturally deal with occlusions. We show that [1] which uses a similar objective also obtains a calibrated distribution, as we predicted, but that our method outperforms it on PA-MPJPE.
>
> **Factors influencing calibration** We agree and are aware that sample-based metrics are not the only “culprit” causing miscalibrated distributions. As shown in [2] various architectural choices can cause miscalibration. [2] shows that model capacity has a negative impact on calibration. We observe a similar phenomenon (Table 1 cGNF vs cGNF xlarge; sec. B.4.). In this study, we provide an analysis of the choice of objective. We show that even when all other aspects are equal (sec. 3.3) sampling-based objectives result in miscalibrated distributions and likelihood-based objectives provide a well-calibrated distribution.
>
> **Compare calibration to other methods** We provide additional comparisons to [1] and [3] (Tables 1 and 2). We show that [3] is miscalibrated. [1] is similarly calibrated to cGNF, as we predicted based on our analysis in section 3. However, our method outperforms [1] in $\operatorname{PA-minMPJPE}$.
>
> **Additional Visualizations** We add additional visualizations showcasing the performance and benefits of the model. We added 1) more examples of posterior samples (Fig. S3) 2) visualizations of the effect of occluding a joint (Fig. S5) 3) showcases of overconfident models which are wrong, while the cGNF captures the effect correctly (Fig. 1 and Fig. S6), and 4) failure case analysis showing that the main cause of failure are overconfident 2D keypoint detections (Fig. S7).
>
> [1] Kolotouros, N., Pavlakos, G., Jayaraman, D., & Daniilidis, K. (2021). Probabilistic Modeling for Human Mesh Recovery. 2021 IEEE/CVF International Conference on Computer Vision (ICCV) , 11585-11594.
>
> [2] Chuan Guo, Geoff Pleiss, Yu Sun, and Kilian Q Weinberger. On calibration of modern neural networks. June 2017.
>
> [3] Tuomas P. Oikarinen, Daniel C. Hannah, and Sohrob Kazerounian. Graphmdn: Leveraging graph structure and deep learning to solve inverse problems, 2020

---

### Decision · Program_Chairs · 2023-01-20

**Decision:**

Reject

**Justification For Why Not Higher Score:**

- soundness of propose ECE metric not clear, does not seem qualified

**Justification For Why Not Lower Score:**

n/a

**Metareview: Summary, Strengths And Weaknesses:**

This paper raises an interesting and overlooked problem for multi-hypothesis 3D human pose estimation.  Specifically, existing approaches tend to yield and favour miscalibrated distributions due to the use of the minMPJPE metric.  To mitigate the mis-calibration, the paper proposes adopting an existing metric (ECE) and couples it with a normalising flow model to learn conditional pose distributions.

All the reviewers found the idea interesting and appreciate the analysis.  However, each reviewer had concerns over the clarity and questions.  Many of the clarity issues were resolved throughout an active discussion period between the reviewers and authors.  However, at the end of the discussion period, some concerns still remained for the reviewers.  The biggest point of contention shared by several reviewers was the soundness of the proposed ECE metric and at the end of the review period, 3 of the 4 authors recommend reject, while a fourth reviewer is borderline.

Having read through the paper and exchange between the reviewers and authors, the AC concurs that the paper needs further work before it is ready for publication.